# Multi-site clonality analysis uncovers pervasive heterogeneity across melanoma metastases

Roy Rabbie [1,2,13], Naser Ansari-Pour [3,13], Oliver Cast [4], Doreen Lau [5], Francis Scott[5], Sarah J. Welsh[2], Christine Parkinson[2], Leila Khoja[6], Luiza Moore [7,8], Mark Tullett[9], Kim Wong [1], Ingrid Ferreira [1], Julia M. Martínez Gómez[10], Mitchell Levesque[10], Ferdia A. Gallagher [5], Alejandro Jiménez-Sánchez[4], Laura Riva[1], Martin L. Miller [4], Kieren Allinson[8], Peter J. Campbell [7], Pippa Corrie[2], David C. Wedge [3,11,12✉] & David J. Adams [1✉]

Metastatic melanoma carries a poor prognosis despite modern systemic therapies. Understanding the evolution of the disease could help inform patient management. Through whole-genome sequencing of 13 melanoma metastases sampled at autopsy from a treatment naïve patient and by leveraging the analytical power of multi-sample analyses, we reveal evidence of diversification among metastatic lineages. UV-induced mutations dominate the trunk, whereas APOBEC-associated mutations are found in the branches of the evolutionary tree. Multi-sample analyses from a further seven patients confirmed that lineage diversification was pervasive, representing an important mode of melanoma dissemination. Our analyses demonstrate that joint analysis of cancer cell fraction estimates across multiple metastases can uncover previously unrecognised levels of tumour heterogeneity and highlight the limitations of inferring heterogeneity from a single biopsy.

[1] Experimental Cancer Genetics, The Wellcome Sanger Institute, Hinxton, Cambridgeshire, UK. [2] Cambridge Cancer Centre, Cambridge University Hospitals NHS Foundation Trust, Cambridge, UK. [3] Big Data Institute, Nuffield Department of Medicine, University of Oxford, Oxford, UK. [4] Cancer Research UK Cambridge Institute, University of Cambridge, Li Ka Shing Centre, Robinson Way, Cambridge, UK. [5] Department of Radiology, School of Clinical Medicine, University of Cambridge, Box 218, Cambridge Biomedical Campus, Cambridge, UK. [6] Institute of Immunology and Immunotherapy, College of Medical and Dental Sciences, Vincent Drive, University of Birmingham, Birmingham, UK. [7] The Cancer, Ageing and Somatic Mutation Programme, Wellcome Sanger Institute, Hinxton, Cambridgeshire, UK. [8] Department of Pathology, Cambridge University Hospitals NHS Foundation Trust, Cambridge, UK. [9] St Richard's Hospital, Spitalfield Lane, Chichester, UK. [10] Department of Dermatology, University of Zurich, University of Zurich Hospital, Gloriastrasse 31, CH-8091 Zurich, Switzerland. [11] Oxford NIHR Biomedical Research Centre, Oxford, UK. [12] Manchester Cancer Research Centre, University of Manchester, Manchester, UK. [13] These authors contributed equally: Roy Rabbie, Naser Ansari-Pour. ✉email: david.wedge@manchester.ac.uk; da1@sanger.ac.uk

Large-scale sequencing studies in cutaneous melanoma have revealed the complex mutational landscape of the disease[1–4]. However, few studies have explored the temporal and spatial evolution of molecular alterations acquired during disease progression. Such findings may inform risk and our understanding of the mode of metastatic spread, with implications for future patient management. Current methods for reconstructing evolution from bulk sequencing data rely on computational approaches to identify sets of mutations that are present in a similar proportion of cells within the tumour[4]. For each set of mutations, the fraction of cancer cells (cancer cell fraction, CCF) carrying them may be estimated from their allele frequencies by adjusting for purity and copy number. Recent studies have used these algorithms to infer the evolutionary relationship between cell populations across multiple samples, correlating these insights with changes in disease progression or therapy response. As neoplastic cells proliferate, some of their daughter cells can acquire mutations that convey a selective advantage, allowing them to become precursors for new tumour cell lineages[5]. In the metastatic context, dissemination of cells from multiple lineages may cause admixtures of cell populations to spread between different metastases, likely with different CCFs at each site. By clustering mutations according to their CCFs across multiple samples simultaneously, it is possible to identify cell populations from the same lineage spread across multiple sites. Further, by comparing these cell populations based on their CCFs across multiple sites simultaneously, it is possible to derive their ancestral relationship. For example, if one cell population is ancestral to another, its CCF must be greater in at least one sample and greater than or equal to the CCF of the descendant cell population in all other samples, when assuming the infinite sites assumption[6]. It should be noted that by constructing trees from clusters of mutations we avoid potentially inaccurate inferences arising from the construction of sample trees when samples are an admixture of cells from multiple lineages[7]. Moreover, joint analysis of CCFs across multiple samples enables the identification of complex intermixtures of cell populations spread across multiple samples from a primary tumour, as well as complex patterns of tumour cell metastasis[8–14]. Other approaches harnessing sophisticated biogeographic models to reconstruct clonal relationships across multiple samples have also provided detailed spatio-temporal insights of tumoural evolution[15].

Throughout this study, we refer to mutations (and mutation clusters) observed in all tumour cells within a sample as 'clonal', those found in a subset of tumour cells as 'subclonal' and those found clonally in all samples from the same patient as 'truncal'. We note that the term 'trunk' is used here in the same sense as the term 'root branch' in the phylogenetic literature. The term 'intratumour heterogeneity' (ITH) has been previously used to refer to heterogeneity identified from single or multi-sampling of tissue from a primary tumour. In this paper we extend the definition of ITH to 'intra-patient tumour heterogeneity', using it to refer to the observation of variants within a tumour that are non-truncal, including variants that may be clonal within some individual samples.

The mutational load of melanoma is one of the highest among all malignancies[16] and, as somatic mutations provide an insight into a cancer's initiation and evolution, genome sequencing studies can provide valuable insights into the progression of the disease. Using targeted panel sequencing of 263 cancer driver genes across 12 primary melanomas matched with regional metastases, Shain and colleagues[17] demonstrated that whilst some primary melanomas and matching regional metastases have pathogenic mutations in just one branch of the phylogenetic tree, there were no driver mutations exclusive to metastases (i.e., not shared with the primaries)[17]. By further showing that most somatic alterations (point mutations and copy number changes) were shared, the authors concluded that primary melanomas and melanoma metastases tend to select for the same set of pathogenic mutations. One feature of such studies is that the clonal composition of each sample is determined using the presence or absence of mutations in each sample. However, this type of modelling also relies on the estimation of clone frequencies, which is vital for the identification of two or more clones per sample and for accurate phylogenetic reconstructions[18]. A recent whole-exome sequencing (WES) study of 86 distant metastases obtained from 53 patients used variant allele frequency (VAF, proportion of reads supporting a mutant allele in parallel sequencing data) of shared vs. private mutations in each lesion to infer the likely clonal status of private mutations within each sample[19]. Although many private mutations were subclonal, this study found polyclonal seeding (defined as a sample harbouring subclonal mutations from 2 or more clonal lineages each of which is also found in another tumour site, thus representing multiple seeding events by two or more genotypically distinct cells[8]) to be a rare event[19]. A picture has therefore emerged whereby the majority of mutations in melanoma metastases are truncal and shared by all progeny. Leading up to the formation of a primary melanoma, a stepwise model of progression has been proposed, which includes selection for particular advantageous molecular alterations (including copy number aberrations), facilitating the sequential transition through successive stages[20,21]. Although this model is well-established for the progression of pre-malignant precursor lesions to invasive primary melanomas[22] the evidence for its ubiquity in metastatic progression is less conclusive.

Multi-site sequencing studies in melanoma have thus far been based on a small number of single nucleotide variants (SNVs) falling in coding exons, with gene panels focussed on SNVs in known cancer genes[17,19,23–26]. While the high depth of sequencing used in these studies enables the detection of rare variants, the number of variants detected will be orders of magnitude lower than that from whole genome sequencing (WGS) and some clonal lineages may therefore go undetected. The VAF can also be affected by contributions from alleles in stroma and infiltrating immune cells, as well as the presence of both the mutated and wildtype alleles in the tumour. Importantly, changes to the copy number of a locus may also alter the VAF dramatically and, if not accounted for, will result in inaccurate clonal frequency estimates, giving a misleading picture of the clonal structure of a tumour[18]. For example, a mutation that has occurred on a chromosome that is subsequently duplicated is carried by two out of three chromosomal copies, whereas a mutation that occurred after the gain is carried by one out of three copies. Indeed, whole-genome duplication and other copy number aberrations have been shown to vary across melanoma metastases from the same patient, evolutionary changes that may not be evident from the analysis of SNVs alone[19]. Inferring clonality from allelic frequencies therefore requires an integrative approach harnessing the most sensitive sequencing technologies, while considering measures of tumour ploidy and purity.

In this study, we present a genome-wide analysis of multiple melanoma metastases sampled at autopsy from a treatment-naïve patient. Using multi-sample clonality analyses across 13 whole-genome sequenced metastases from this patient, as well as multi-site analyses of whole-exome sequenced metastases from a further 7 patients, we identify clusters of co-occurring truncal, clonal and subclonal mutations across multiple samples, and uncover the chronological order of genomic alterations. We show that metastases in different organs may have distinct clonal lineages and reveal that melanoma metastases harbour previously unrecognised levels of ITH.

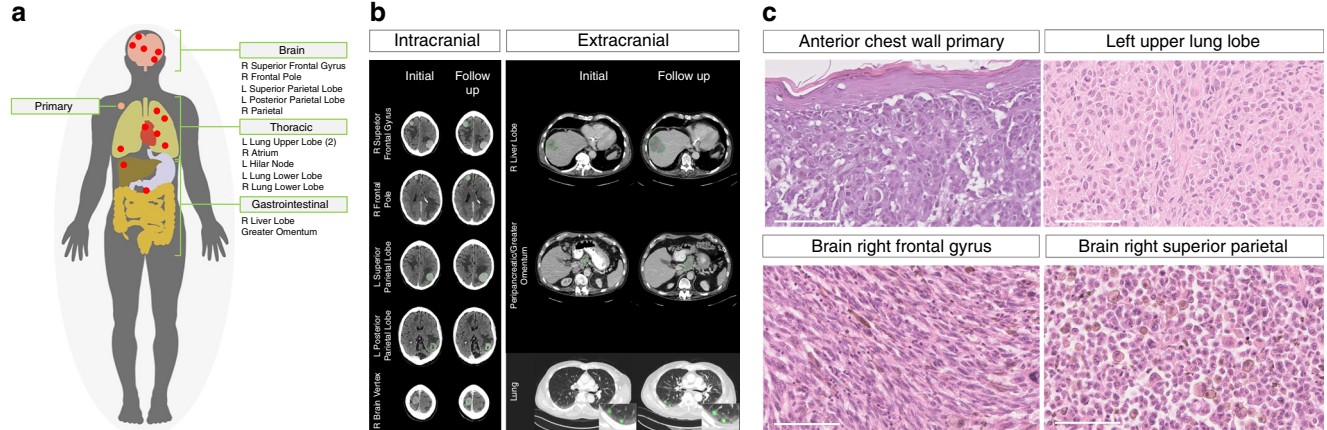

**Fig. 1 Clinical presentation and sequalae of index autopsy case. a** Sites of 13 metastases sampled during the autopsy and the anterior chest wall cutaneous primary melanoma. **b** Axial CT imaging from the brain, chest and abdomen before (left) and after (right) whole brain radiotherapy (imaging 5 weeks apart). Brain CT images represent the following metastatic sites (from top to bottom); right superior frontal gyrus, right frontal pole, left superior parietal, left posterior parietal and right brain vertex (the latter corresponding to the 'right parietal' sample labelled in the remainder of the text). Chest/abdomen CT images represent the following metastatic sites (from top to bottom); right lobe of liver, peripancreatic/greater omentum (corresponding to the 'greater omentum' sample labelled in the remainder of the text) and right lower lung. **c** Histological analyses (Hematoxylin-eosin images), from top left to bottom right; primary melanoma from the anterior chest wall, distant metastasis from; the left upper lobe of the lung, right frontal gyrus of the brain and right superior parietal lobe of the brain. Morphological appearances differ across the tumours, including varying cellular morphology and pigmentation. One representative region from each tumour sample is shown. The thick white line represents 100 µm in 40-fold magnification.

## Results

**A clinical course characterised by multi-organ metastases**. Our index case was a 71-year-old male of European descent with no relevant family history, who initially presented with a 1.2 mm Breslow thickness non-ulcerated, Clark level 3 superficial spreading melanoma which was resected from the anterior chest wall with a wide local excision (Fig. 1a). The patient declined a sentinel lymph node biopsy and staging scans were clear for distant metastases. Five years later, the patient presented to the emergency department with sudden onset receptive dysphasia and dyspraxia. A contrast computerised tomography (CT) head scan showed multiple enhancing lesions in both cerebral hemispheres with adjacent vasogenic oedema consistent with metastases (Fig. 1b). A staging contrast CT also showed multiple lung, liver and retroperitoneal lymph node metastases (Fig. 1b). A biopsy of one of the liver lesions confirmed metastatic melanoma. Mutation-specific immunohistochemistry for $BRAF^{V600E}$ did not detect the mutated protein, but subsequent targeted panel sequencing identified an activating $BRAF^{V600R}$ mutation. However, in view of his poor overall performance status and on discussion with the patient and his family, he chose to be managed with best supportive care. The patient consented to undergo a research autopsy as part of the ethically-approved MelResist study (see 'Methods' section). He received corticosteroids with marked improvement in neurological symptoms and underwent wholebrain radiotherapy 30 Gy in 10 fractions. A repeat staging CT scan 2 weeks after completing radiotherapy revealed stable brain metastases but widespread progression of the extracranial disease (Fig. 1b and Supplementary Fig. 1). He died four weeks later.

During a research autopsy, metastases were identified macroscopically in the brain, lung, liver and retroperitoneum, as well as the right atrium, the latter was identified as an 18 mm polypoid lesion arising from the endocardial surface. The cause of death was identified as a saddle pulmonary embolus. In total, 13 metastases were sampled at autopsy and a further 2 samples were obtained from the archived anterior chest wall primary melanoma for further molecular analyses (Fig. 1 and Supplementary Table 1). Histopathological analyses of the metastases confirmed metastatic melanoma. Morphological heterogeneity was observed between metastases based on cellular features (ranging from epithelioid to spindle cell), as well as in the degrees of pigmentation and necrosis (Fig. 1c).

**Melanomas are dominated by UV-induced clonal mutations**. Whole genome sequencing of the 13 metastatic tumours sampled at autopsy, which were sequenced to a median depth of 38x, revealed a union list of ~118,000 SNVs. We detected 1993 putative somatic indels, of which 10 were frameshifts and common to all metastases (see 'Data availability' section). All 13 metastases carried an activating missense $BRAF^{V600E}$ mutation (c.1798_1799delGTinAG), as well as mutations in the melanoma driver genes $PTEN^{A43T}$ and $MAP2K1^{G128S}$ (the latter has not been previously reported in the COSMIC database[27]), and a splice-site variant in $ARID2$, all of which were truncal across all metastases. We further explored the clonal architecture using the Cancer Cell Fraction (CCF), determined by adjusting the variant allele frequencies of SNVs for copy number aberration (CNA) status and the extent of normal cell contamination (i.e., purity)[4]. Briefly, multidimensional Bayesian Dirichlet Process-based mutation clustering (ndDPClust[4,28]) was used to identify truncal, clonal and subclonal mutation clusters based on the CCF of the union list of somatic SNVs across all 13 metastases. Given that clustering analyses were initially undertaken on the metastatic tumours, subsequent references to metastatic-truncal and metastatic non-truncal mutation clusters apply only to those identified in the metastases of the index autopsy case. Using this approach, we found that >90% of all somatic variants in the metastases were metastatic-truncal, with only one additional cluster which represents at least 1% of the SNVs ($N = 1651$, 1.35%). The large metastatic-truncal cluster was dominated by C > T transitions at dipyrimidines (characteristic of UV-induced mutational damage[29]) and was shared across all metastases, implying that ITH was absent (Fig. 2a). We next filtered for artefactual clusters and SNVs within regions of ambiguous copy number status across all samples (see 'Methods' section) and subtracted variants assigned to the major metastatic-truncal cluster which uncovered a union list of 2247 unique non-truncal variants (Fig. 2b). Overall, 22/2247 of these variants fell within the

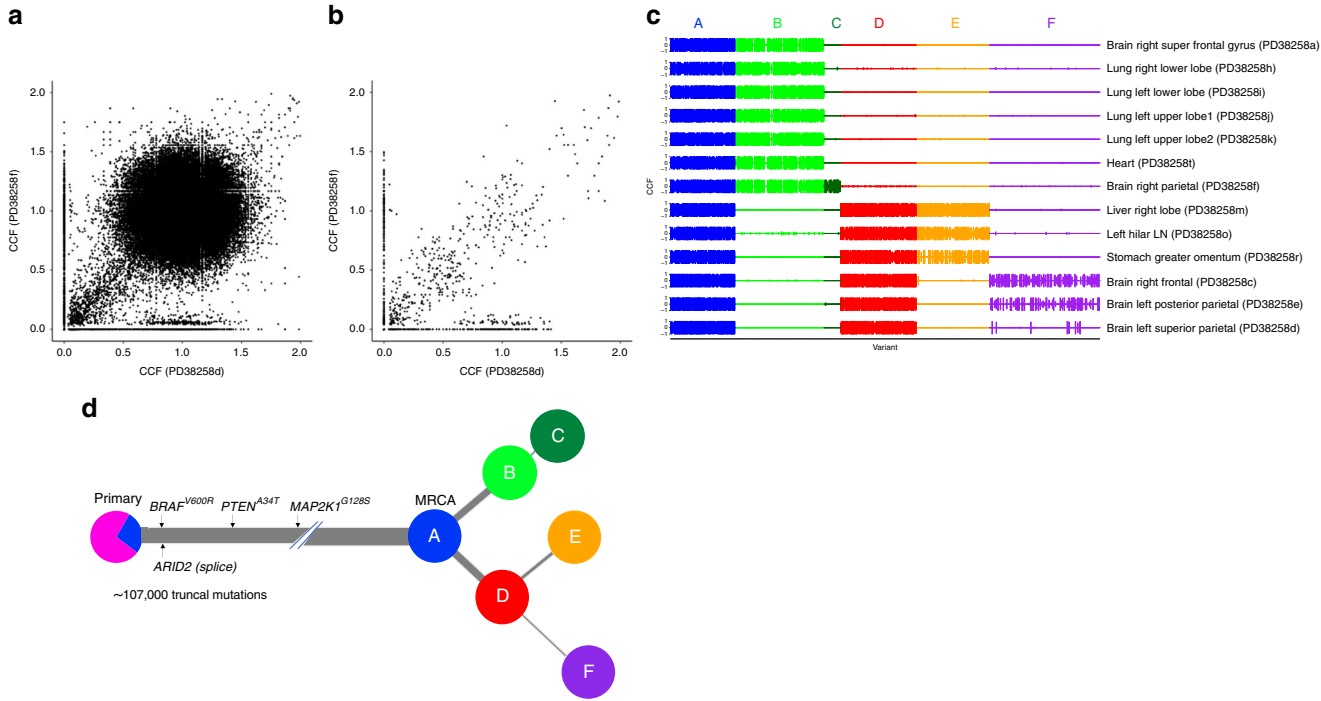

**Fig. 2 Subtracting the clonal cluster revealed subclonal diversification in the index autopsy case. a** Point estimates of the cancer cell fraction (CCF) are represented here as black dots across two representative samples in a density plot (left super parietal (PD38258d) and right parietal brain metastases (PD38258f), represented across the X and Y axes, respectively). We observed a large cluster of mutations at (1,1), corresponding to single nucleotide variants (SNVs) present in all the cells in both sites (CCF = 1), indicating truncal variants. **b** After removing the metastatic truncal variants, the metastatic non-truncal mutation clusters uncovered subclonal mutations. Although there are still a small number of SNVs at CCF 1, these did-not belong to the metastatic truncal cluster of mutations, representing those mutations found clonally in all metastases. **c** The clusters of metastatic non-truncal mutations are represented in a CCF distribution plot, wherein rows reflect samples and columns reflect alphabetically and colour-assigned mutation clusters (n = 6). The x-axis represents the SNVs within each cluster and the y-axes represent CCF; the latter has been plotted in positive and negative directions for improved visualisation of clusters. **d** Alphabetically and colour-assigned mutation clusters from **c** are represented in circles, where branch length is proportionate to the number of SNVs in the cluster (filtering SNVs occurring in regions with different CNA status across samples reduced 118,000 SNVs to 107,000 truncal SNVs indicated here) and branch thickness is proportionate to the average CCF of the cluster (across all the samples). Mutations that have occurred before the most recent common ancestor (MRCA) are carried by all tumour cells and define the metastatic truncal cluster of mutations. Truncal mutations in the melanoma driver genes $BRAF^{V600R}$, $PTEN^{A34T}$, $MAPK2K1^{G128S}$ (the latter represents a mutation not previously reported in the COSMIC database[27]) as well as an ARID2 splice-site variant are indicated on the trunk of the tree (the order of the driver mutations displayed on the trunk is arbitrary). The primary tumour colour shading represents the mutation clusters detected in the primary (using targeted sequencing), whereby the blue represents the subclonal mutation cluster within the primary (at average CCF of 0.27), which became fixed and truncal across all metastases.

protein-coding region of the genome of which 14 were protein-altering (all missense). None of these variants were in established cancer driver genes listed in the Catalogue of Somatic Mutations In Cancer (COSMIC)[27]. We undertook a validation experiment by custom capture pull-down sequencing of protein-coding metastatic-truncal SNVs present across all 13 metastases (selected as either cancer driver or loss-of-function SNVs, N = 652), as well as all 2247 metastatic non-truncal SNVs. In this way, 99% of metastatic-truncal SNVs and 92% of all the metastatic non-truncal SNVs were observed to be true variants, instilling confidence in the downstream phylogenetic reconstructions based on these SNVs (see 'Methods' section).

**Mutational cluster analyses reveal distinct clonal lineages**. Applying ndDPClust to the metastatic non-truncal variants from the index autopsy case revealed 6 distinct mutation clusters with variable distribution across all metastases (Fig. 2c). Assessing the distribution of these clusters, as well as the CCF distribution within each cluster across the metastases, we were able to reconstruct a phylogenetic tree (see 'Methods' section for further details). The mutation clusters showed clear lineage separation, such that samples harbouring clusters from one lineage were

mutually exclusive from samples harbouring clusters in the opposing lineage, supporting a clear bifurcation after the most recent common ancestor (MRCA) (Fig. 2c). In particular, cluster B (light green) was present (and clonal) in the first 7 samples belonging to the first clonal lineage and absent in the latter 6 samples belonging to the second clonal lineage, which in-turn were represented by cluster D (red). These mutually exclusive clonal clusters at the first bifurcation suggest that alternative tree solutions are very unlikely (Fig. 2d).

We next assessed the distribution of mutation clusters per sample in order to reconstruct sample-level phylogenetic trees for the metastases (Supplementary Fig. 2). Sample-level trees represent subtrees of the overall phylogenetic tree including just those clones seen within each metastasis, however, in doing this we were able to segregate the samples based on their respective clonal lineage. Analysis in this way revealed two clear lineages, representing distinct waves of metastatic seeding. Interestingly, the brain metastases were represented on both lineages suggesting there were at least two waves of spread to the central nervous system, whereas the lung metastases were derived from a single lineage (Supplementary Fig. 2). In summary, we found that mutation clusters from multiple distinct clones were present across multiple metastatic tumours. This approach has revealed

convincing evidence of polyclonal seeding in metastatic melanoma. This finding however could not have been otherwise resolved had it not been for the removal of the dominant cluster of truncal variants, which masked this complex phylogenetic architecture.

**Metastatic truncal mutations are subclonal in the primary**. We next analysed the representation of metastatic SNVs within two tissue blocks from the original cutaneous primary of the index case resected five years earlier, with the aim of tracing back the ancestral clones. We used targeted panel sequencing with a median coverage of 40×, to ascertain whether selected SNVs identified in the genome-sequenced metastatic tumours were also present in the primary, requiring at least 2 supporting reads reporting the alternative allele to call an SNV in the primary (see 'Methods' section). We found that 573/652 (88%) of the selected metastatic truncal variants could also be detected in the primary, whereas only 1 of the metastatic non-truncal cluster variants was identified in the primary at this depth (see 'Methods' section). By selecting the 144/652 metastatic truncal SNVs present in diploid regions, we were not only able to estimate purity of the two primary samples based on VAF density of SNVs (see 'Methods' section), we were also able to run ndDPClust on both primary tumour samples by assuming diploidy in SNVs within the primary tumours. In addition to the main clonal cluster, we further identified that the two primary tumour samples harboured the same subclone represented by 37 SNVs (at CCF 0.25 95% CI 0.22–0.37 and 0.29 95% CI 0.18–0.34 for samples PD38258u and PD38258v, respectively). No known drivers were uniquely present in this primary tumour subclone. However, we did identify a nonsense variant in *IL1R1*, a gene which is thought to act as a tumour suppressor[30]. This subclonal cluster within the primary tumour might correspond to the lineage that originated the metastases (Fig. 2d). However, the small number of evidential variants warrants further studies in both primary and metastatic melanomas.

**Lineage diversification from analyses of 7 further patients**. In order to assess whether lineage diversification was detectable in further cases, we undertook whole-exome sequencing (WES) of 19 melanoma metastases matched with germline blood samples from an additional 7 patients with metastatic melanoma who had consented to take part in the MelResist study. All samples were obtained from clinically or radiologically progressing disease sites, either at the time of first distant relapse, or following systemic therapy with MAP-kinase directed therapies or immune checkpoint inhibitors (Supplementary Table 2). We identified an average of 598 (range of 108–2088) non-synonymous coding variants (including missense, nonsense and splice-region mutations), and 7 (range 2–15) frameshift variants per patient, both totalled across all samples within each patient (see 'Data availability' section). Six out of 7 patients had metastases arising from a cutaneous primary and 1 patient (MultiSite_WES_Patient1) had an acral primary melanoma. Their metastases (MultiSite_WES_Patient1), as expected, carried a particularly low number of SNVs (only 298 non-synonymous coding variants totalled across metastases). All 7 patients had metastases harbouring an activating *BRAF*$^{V600E}$ driver mutation and, in accordance with previous reports[17,19], all melanoma drivers were represented on the trunks (rather than the branches) of the phylogenetic trees (except for a previously unreported *TP53*$^{R141C}$ mutation in patient MultiSite_WES_Patient1) (Fig. 3). We again used ndDPClust to cluster SNVs according to their respective CCFs (see 'Methods' section). We identified 2–10 distinct clusters per patient with clear evidence of lineage diversification across 6

out of 7 patients, evidenced by the presence of mutation clusters in mutually exclusive subsets of samples (Supplementary Fig. 3). By reconstructing sample-level phylogenetic trees, we identified distinct clonal lineages within each patient and found evidence of polyclonal seeding in two patients (Supplementary Fig. 4). Given that lineage diversification was detected in 6 out of 7 cases (including the acral melanoma patient) based on WES, which has a much lower genomic resolution than WGS, and with as little as two samples per patient in most cases (Supplementary Table 2), we provide strong evidence that ITH is likely to be pervasive in melanoma metastases.

**Subclonal APOBEC signature mutations**. We extracted mutational signatures from the 13 whole-genome sequenced metastases collected from the index case[31]. As expected, all samples were dominated by signature 7 reflecting UV-induced mutagenesis (Supplementary Fig. 5). Within the pool of 2247 non-truncal mutations, however, we found evidence of non-UV induced mutational signatures, including signatures 2 and 13, which represent the action of the APOBEC family of cytidine deaminases (which enzymatically modify single-stranded DNA)[32] (Supplementary Fig. 6A). We found that whilst signature 7 dominated the truncal cluster, it was absent from the branches of the evolutionary tree, which were characterised by the APOBEC mutational signatures suggesting this process might be implicated in later stages of clonal evolution (Supplementary Fig. 6B). Interestingly skin cancer has been shown to have the fifth highest *APOBEC3B* expression rank[33]. However, the dipyrimidine-focused C-to-T mutation pattern of UV eclipses an *APOBEC3B* deamination signature, which we have only uncovered here by separating out the truncal mutations.

**Gene expression analyses reveal clustering within organs**. Gene expression analyses of 11 metastases from the index autopsy case further revealed differences between metastases found across different organs, with principal component analysis (PCA) separating metastases sampled from the brain and lung (Fig. 4a, Supplementary Fig. 7A, B). Interestingly, metastases seeding within the brain clustered together by PCA, despite phylogenetic inferences indicating these likely emanated from differing lineages (Supplementary Fig. 2). In order to identify the tumour-specific genes and biological processes uniquely associated with brain metastases in this patient and mitigate for any potential influence of cellular contamination from the surrounding stromal/immune cells (Fig. 4a), of which purity analyses from copy number calls suggested were <15% (see Supplementary Table 1), we intersected the genes (Fig. 4b) and pathways (Fig. 4c) differentially expressed between both brain metastases ($n = 5$) vs. normal tissue (from the patients' normal brain and lung tissue) (Supplementary Fig. 7C), with those between brain ($n = 5$) vs. lung metastases ($n = 4$) (Supplementary Fig. 7D). The gene *PLEKHA5* was significantly upregulated (log-fold change 4.5, FDR-adjusted *p*-value < 0.003) in brain vs. lung metastases, as well as in brain metastases vs. normal tissues (log-fold change 5.3, FDR-adjusted *p*-value < 0.004). This guanine nucleotide exchange factor has previously been shown to be upregulated in a cell line model of melanoma brain metastasis (cerebrotropic A375Br cells vs. parental A375P cells) and silencing of *PLEKHA5* expression decreased in-vitro potential of these cells to cross the blood-brain barrier[34]. Gene set enrichment analyses also showed significant enrichment of the oxidative phosphorylation KEGG pathway in both brain vs. lung metastases (normalised enrichment score 4.65, FDR-adjusted *p*-value < 0.0001) and in brain metastases vs. normal tissues (normalised enrichment score 3.17, FDR-adjusted *p*-value < 0.0001) (Fig. 4c). This is consistent with recent analyses implicating the upregulation of oxidative phosphorylation

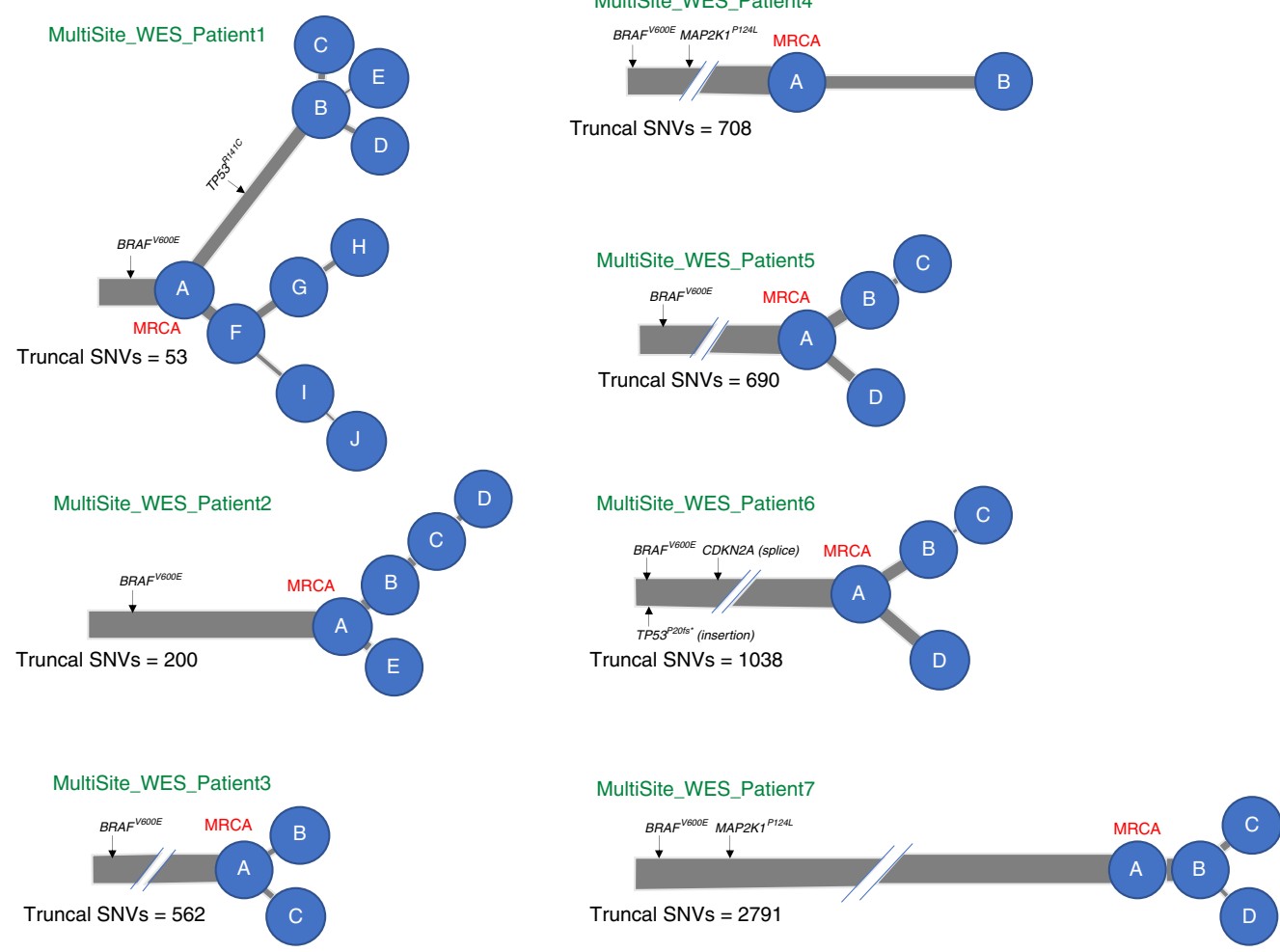

**Fig. 3 Multi-dimensional clustering across metastases from 7 patients uncovers divergent lineages.** Phylogenetic trees of 7 metastatic melanoma patients are indicated, with branch lengths proportional to the number of SNVs within the mutation cluster and branch thickness proportional to the average CCF of the cluster (across all the samples within each patient). The number of truncal SNVs is indicated underneath the respective phylogenetic trees, most recent common ancestor (MRCA) is labelled in red. All patients harboured an activating $BRAF^{V600E}$ driver mutation. In keeping with previous reports[17,19], all other melanoma drivers were represented on the trunk (rather than the branches) of the phylogenetic trees, except for one patient (MultiSite_WES_Patient1) harbouring a previously unreported $TP53^{R141C}$ mutation which was unique to one branch and completely absent in the three samples on the opposing branch (despite adequate coverage, >40x).

in patient-matched brain vs. extracranial metastases, as well as further functional studies demonstrating that inhibition of this pathway resulted in increased survival in both implantation xenografts and spontaneous murine models of melanoma brain metastases[35].

Immune cell estimation of bulk tumoural mRNA from the above mentioned 11 metastases (as previously described[36]), further revealed evidence of distinct tumour-immune microenvironments (Fig. 5). The brain metastases in particular had relatively few immune cells compared to lung and other extracranial metastases, which might corroborate studies suggesting the brain represents a relatively immuno-privileged organ[35,37]. Inflammatory macrometastases in the brain expressed high levels of transcripts for activated M2 macrophages (including significant upregulation of the macrophage marker *CD163* in brain vs. lung metastases, log-fold change 5.5, FDR-adjusted *p*-value < 0.004) which have been described as having anti-inflammatory or tumour supporting activities, including in malignant brain tumours (although it is important to recognise that it may be difficult to distinguish between microglia and

macrophages, both of monocyte lineage, using these methods)[38]. In summary, therefore, despite having overall similar mutational landscapes, we identified distinct tumour-immune microenvironments by gene expression analyses, which likely reflects differences in the tumour microenvironment ecosystem[39].

### Discussion

Although melanoma is associated with a large number of somatic SNVs, the genomic diversity of melanoma metastases has previously been reported to be low, with most SNVs expected to be shared across tumours[17]. Recently, the International Cancer Genome Consortium Pan-Cancer Analysis of Whole Genomes (PCAWG) initiative[40] leveraged WGS data to infer evolutionary relationships across multiple cancer types, further showing that metastatic melanomas may be monophyletic, with a single clone appearing to seed metastases and, when compared with other cancers, may uniquely lack ITH[41]. In our study, analyses of clonal structure from multi-site whole-genome sequenced melanoma metastases provided a powerful method to detect mutation clusters and a unique insight into clonal evolution. In agreement

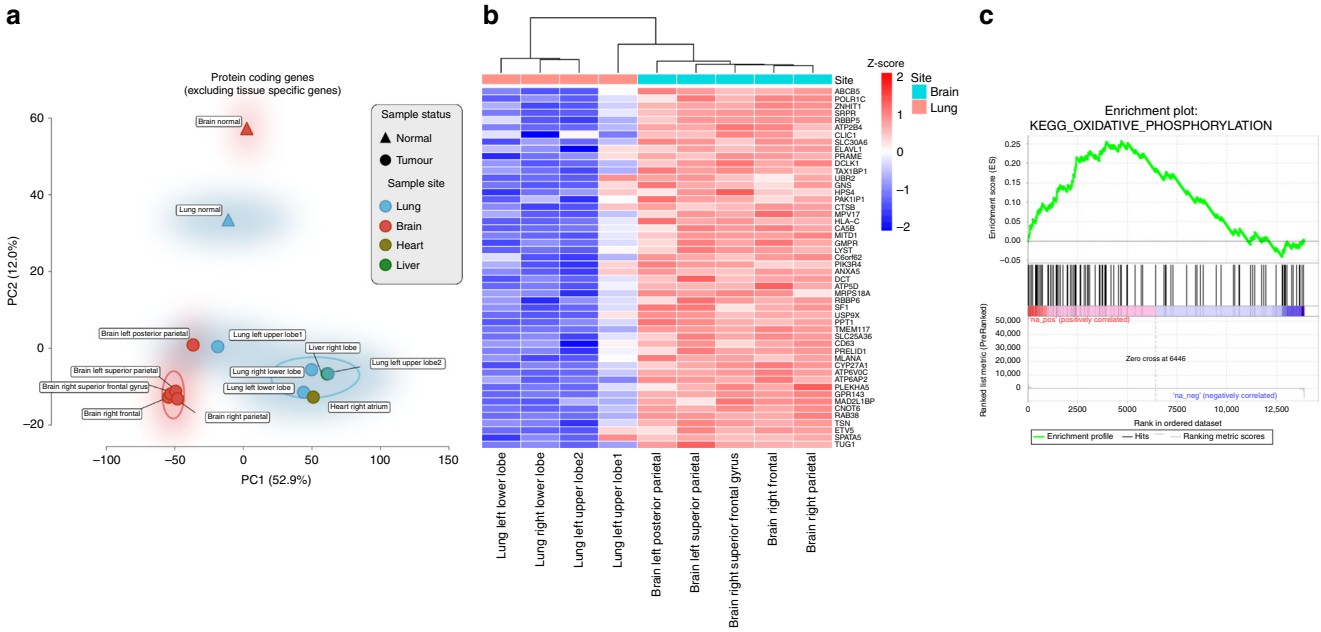

**Fig. 4 Gene expression analyses reveals regional separation of site-specific metastases. a** Principal component analysis of protein-coding gene expression across all samples. A regional separation can be seen between the brain ($n = 5$, coloured red circles) and lung metastases ($n = 4$, coloured blue circles) on PC1 along the x-axis (these samples are circled using a kernel density estimation), which accounted for greater variance than the separation between the tumour ($n = 11$) and normal samples ($n = 2$) by PC2 along the y-axis (53% vs. 12%, respectively), indicating that expression patterns are at least partially influenced by cellular contamination from the surrounding stromal cells. **b** Heatmap showing the top 50 intersecting genes that are differentially expressed between both brain vs. lung metastases and brain metastases vs. normal tissue comparisons (see "Methods" section). Z-score scale indicates normalised gene expression and represents the number of standard deviations away from the mean (red-blue denoting high-low normalised gene expression, respectively). **c** Pre-ranked gene set enrichment analysis of genes associated with metastases to the brain revealed biological pathways that might have been implicated in brain metastases. The enrichment plot provides a graphical view of the enrichment score for a gene set. The top portion of the plot shows the running enrichment score for the gene set as the analysis walks down the ranked list. The middle portion of the plot shows where the members of the gene set appear in the ranked list of genes and the bottom portion of the plot shows the value of the ranking metric as you move down the list of ranked genes. Oxidative phosphorylation was the most statistically significant over expressed MSigDB KEGG pathway, enriched in both the brain vs. normal tissue and brain vs. lung comparisons (FDR corrected p-value < 0.0001, calculated by constructing a histogram corresponding to null enrichment scores see 'Methods' section), and has recently been linked to melanoma brain metastases in both human and murine analyses[35].

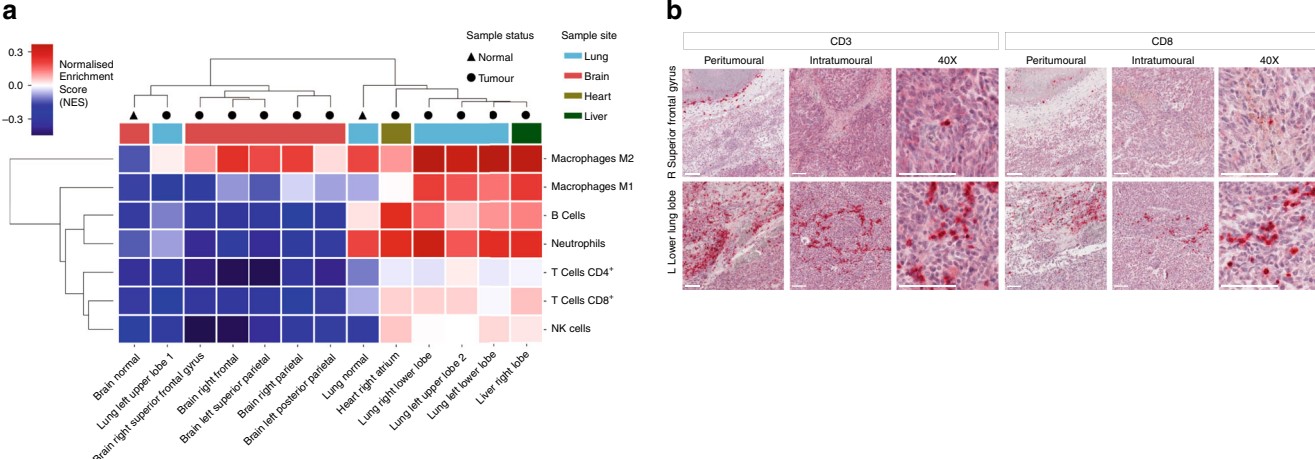

**Fig. 5 Regional differences in immune cell representation; brain vs. lung metastases. a** Immune cell deconvolution of mRNA using hierarchical clustering of Consensus[TME] scores[66]. Scale indicates high-low normalised enrichment scores (red-blue). Both the brain normal and metastases were relatively immuno-privileged when compared to lung normal and lung metastases (note that one of the four lung metastases, left upper lobe 1, was an outlier clustering with the immune-sheltered brain samples). **b** Immunohistochemical (IHC) validation against CD3 and CD8 in a representative brain and lung metastasis. One representative region from each tumour sample is shown. Chromogen red indicates positive immune cell staining. The thick white line represents 100 μm in 10-fold (peritumoural and intratumoural) and 40-fold magnification, respectively.

with previous literature, we identified a single cluster of truncal variants ubiquitously represented across all metastases and representing >90% of all somatic SNVs. Metastatic truncal variants dominated downstream phylogenetic reconstruction analyses. Initial analyses therefore showed that ITH appeared to be absent and that metastases were derived from a single parental clone harbouring the majority of genetic alterations. However, by subtracting this dominant cluster of variants, we were able to identify non-truncal clones and subclones. Assessment of the representation of these clones across the metastatic tumours revealed the ancestral relationships between metastases and uncovered a phylogenetic structure that reframes currently accepted models of metastatic dissemination. In particular, these patient-matched primary and metastatic tumours provided the power to detect SNVs that were present only in a subset of sequenced samples, thereby increasing the power of these reconstruction approaches.

We found evidence of lineage diversification across metastatic melanoma exomes from a further 6 out of 7 patients (including in one case of metastases from an acral primary), indicating that, even with a much lower sequencing breadth, and nearly two orders of magnitude fewer SNVs (relative to whole-genome sequencing), detailed clonal lineages could still be inferred, and pervasive ITH was observed. The detection of ITH using lower-resolution WES from archival formalin-fixed paraffin embedded (FFPE)-derived samples is particularly relevant to clinical practice, where the majority of samples are still stored in paraffin, and where custom pull-down is much more readily available than whole-genome sequencing approaches[42]. It is therefore our impression that previous studies suggesting that melanoma metastases lack heterogeneity may have been confounded either by the use of VAF as a surrogate for CCF[43], or by the lack of power to separate subclones through single sample analyses[41] (rather than by the limits of resolution of targeted sequencing approaches). Our analyses should therefore serve as a cautionary tale in future phylogenetic analyses that still define trunk and branch mutations by the presence or absence of shared variants and that do not consider CCF calculations (adjusting somatic VAFs with tumour purity and CNA status). In a previous single-patient WGS study analysing a primary acral melanoma and its concurrent ipsilateral inguinal lymph node, a wide spectrum of SNVs and copy number alterations were found to be shared between the primary and metastatic tumour, however, the phylogenetic architecture could not be fully reconstructed[44]. By harnessing the power of CCF calculations across 6 metastases from our acral melanoma patient, we identified divergent lineages. A recent detailed multi-regional clonality analysis in uveal melanomas has also found multiple driver mutations in the branches of the phylogenetic trees, suggesting that these melanomas also continue to evolve as they progress from primary to metastatic disease[45]. Therefore, we postulate that multi-sample analyses may reveal that ITH is characteristic of other melanoma subtypes, although further studies in these rarer subtypes are warranted.

Analysing skin/subcutaneous metastases in 8 patients with cutaneous melanoma, Sanborn and colleagues previously showed that locoregional relapses arose from different cellular subpopulations of the primary tumour[46]. Our analyses support these findings, and show that lineage diversification is associated with both locoregional, as well as more distant metastatic spread. The phylogenetic trees were dominated by long trunks, with smaller branches representing subclonal diversification (palm tree resemblance) (Fig. 2d). Driver mutations generally arose before subclonal diversification and were found primarily on the long trunks of the trees (Figs. 2d and 3). This contrasts with recent reports in prostate cancer[11], where branching generally occurred throughout the tumours' evolutionary trajectories, and with studies of various other tumour types reporting the frequent occurrence of subclonal driver mutations, but is concordant with previous studies of melanoma[17,19]. Interestingly, a single subclonal mutation cluster (cluster F, shown in purple in Supplementary Fig. 2) was found subclonally in brain metastases from the index autopsy case. However, in keeping with previous analyses[19,46,47], polyclonal seeding was generally a rare event in this cohort (Supplementary Figs. 2 and 4). Although the index patient underwent whole-brain radiotherapy six weeks prior to the autopsy, we did not detect differences in mutational signatures between the brain and extracranial metastases[31,48] (see 'Data availability' section), while the lack of prior systemic therapy further supports the mutational processes being reflective of evolutionary changes during dissemination, rather than the result of treatment.

The catalogue of somatic mutations in cancer is the aggregate outcome of exposure to one or more mutational processes. Each process generates mutations characterised by a specific combination of nucleotide changes and nucleotide contexts, therefore providing a signature that can be used for its identification[31,49]. Studies by Shain and colleagues suggested that UV is the dominant factor associated with the initiation of precursor lesions and dominates every stage of tumour evolution, from the progression of pre-malignant lesions to primary melanoma and to metastases[20,21]. Interestingly however, other studies have reported a reduction of the proportion of mutations associated with the UV-induced mutational signature in branch (non-truncal) mutations, suggesting that mutations arising later in melanoma progression may occur as a result of increased activity of other mutational processes[17,19]. This is consistent with lung cancer analyses, where subclonal lineages acquired mutations that lacked the tobacco-smoking signature and were replaced with mutations associated with APOBEC cytidine deaminase activity[9]. Consistent with these analyses, we found that whilst the truncal cluster of mutations in the autopsy index case was dominated by signature 7, this was absent in the subclonal mutation clusters which also appeared to be replaced by APOBEC-associated mutations. Further studies will be required to ascertain whether these observations can unravel new biological insights.

In summary, through leveraging the power of clonality analyses across multiple whole-genomes we were able to identify rich clonal architectures and uncover ITH of melanoma metastases obtained at autopsy of a single patient, a structure which would not have been evident through single-site reconstructions. Using the same approach, we found further evidence of divergent lineages in whole-exome sequenced metastases obtained from 6 out of 7 additional melanoma patients, one of which was an acral melanoma, suggesting that this is independent of sequencing breadth or depth. Our ability to detect distinct clonal lineages was greatly enhanced by leveraging the power from multiple samples and uncovers conclusive evidence of ITH in melanoma metastases. Future large-scale studies incorporating clonal analyses across multiple metastases will be required to further delineate how these tumours evolve, and provide insights into whether interrupting this process could contribute to patient management.

## Methods

**Patient enrolment**. All patients were recruited to the MelResisist prospective non-interventional study sponsored by Cambridge University Hospitals NHS Foundation Trust. The study was approved by the National Research Ethics Committee (NREC) North East on the 17th October 2011 IRAS project ID 66161 and REC reference 11/NE/0312. All patients provided written informed consent to take part. All cases were also ethically approved by the Sanger Institute's human materials and data management committee. The research autopsy was conducted 48 h after the patient's death, during which time the body had been stored at 4 °C. Sixteen 1cm-diameter core biopsies were sampled from the centre of each metastatic

tumour and snap frozen in liquid nitrogen at −80 °C. These were used for the extraction of bulk DNA and RNA, as well as for the creation of stained H&E slides for direct histopathological assessment of the sequenced regions. The remaining multi-site exome-sequenced cases were also identified from the prospective Mel-Resist trial, and were selected based on their availability of banked multi-site metastases for molecular interrogation. A total of 7 patients with 21 metastases (median 2, range 2–6 metastases per patient) were identified for this analysis. All samples and clinical details are listed in the Supplementary Tables 1 and 2.

**Extraction and quality assessment of DNA and RNA**. Histopathological assessments were performed by two consultant histopathologists (KA and MT), who confirmed that all tumours were composed of >90% neoplastic cells. Macro-dissection of fresh tumour cores from the autopsy case was performed with a sterile scalpel. DNA and RNA were extracted from the fresh tumour cores using the AllPrep combined DNA/RNA Mini Kit (Qiagen Ltd.) according to manufacturer's recommendations. All the multi-site exome-sequenced cases were obtained as 1.0 mm diameter cores micro-dissected from the original FFPE block. Genomic DNA was extracted from the FFPE cores using the QIAamp FFPE Tissue kit from Qiagen according to manufacturer's instructions. Germline DNA was extracted from peripheral blood mononuclear cells collected before death from all cases, using the DNeasy Blood and Tissue Kit (Qiagen). To confirm that the tumours and germline DNA were derived from the same patient Fluidigm genotyping was performed. All DNA samples were quantified using the PicoGreen dsDNA Quantification Reagent according to manufacturer's recommendations (Invitrogen). The structural integrity of DNA was checked by gel electrophoresis. RNA quantity and quality were assessed using Agilent's 2100 bioanalyzer.

**Laser capture microdissection of the cutaneous primary from the autopsy case**. Two FFPE tissue blocks from the index autopsy patient's archival primary tumour (cutaneous melanoma from the anterior chest wall, samples PD38258u and PD38258v) were processed into 5 μm histology sections, deparaffinised with ethanol thrice and stained with Gill's haematoxylin for 20 s. Malignant melanocytes from each section were isolated by a histopathologist (LM) using laser capture microdissection and collected in separate Eppendorf tubes. These were then lysed with lysis buffer ATL and digested with proteinase k (Qiagen Ltd.). Extraction of nucleic acids was performed using the QIAmp DNA FFPE extraction kit (Qiagen Ltd.) according to manufacturer's recommendation.

**Whole genome sequencing and somatic variant detection**. Paired-end sequencing of the metastatic tumours and matched normal was performed on the Illumina X10 platform at the Wellcome Trust Sanger Institute to generate 150 base-pair reads. Sequencing reads were aligned using BWA-MEM (v0.7.12)[50] to the human reference genome (NCBI build GRCh37). The resulting sequencing coverage ranged from 33-fold to 43-fold (median 38-fold). Caveman (v1.11.2)[51] and Pindel (v2.2.4)[52] were used to call somatic SNVs and indels, respectively. The minimum base quality score for somatic variant calling was set to Phred 30. ANNOVAR[53] was used to predict the effect of variants on genes and to assign rsIDs for known variants based on dbSNP Human Build 150. The alignments for all variants are reported in the 'Data availability' section. To call rearrangements we applied the BRASS (breakpoint via assembly) algorithm, which identifies rearrangements by grouping discordant read pairs that point to the same breakpoint event (github.com/cancerit/BRASS). BRASS rearrangements were used to search for balanced inversions, which have been previously associated with radiation-induced mutagenesis[48], and were particularly relevant to explore in the radiotherapy-treated brain metastases from the index case.

**Whole exome-sequencing of multi-site metastases cases**. Exome capture was performed using Agilent's SureSelect bait. Paired-end sequencing was performed using the Illumina HiSeq platform at the Wellcome Trust Sanger Institute to generate 75 bp reads. Sequencing reads were aligned using BWA-MEM (v0.7.12)[50] to the human reference genome GRCh37. PCR duplicates, secondary read alignments, and reads that failed Illumina chastity (purity) filtering were flagged and removed prior to running variant and copy number calling. The resulting sequencing coverage after filtering ranged from 33-fold to 95-fold (median 52-fold) in the tumoural samples and median 54-fold across the germline blood samples. Caveman (v1.11.2)[51] and PINDEL (v2.2.4)[52] were used to call somatic SNVs and indels, respectively. The minimum base quality score for somatic and germline variant calling was set to Phred 30. ANNOVAR[53] was used to annotate SNVs (based on Caveman) and indels (based on PINDEL) for functional classification and to assign rsIDs for known variants based on dbSNP Human Build 150 (see 'Data availability' section).

**Copy number aberration (CNA) profiling**. Segmental copy number information was derived for each of the 13 metastatic tumours using the Battenberg algorithm (v3.2.2)[10]. This was also used to estimate tumour cellularity and ploidy, and calculate allele-specific copy number profiles[12]. Sequenza (v2.1.2)[54] was used to estimate tumour cellularity and ploidy from the tumour-normal pairs in the multi-site FFPE-extracted exome sequenced cohort, as well as to calculate allele-specific copy number profiles (see 'Data availability' section). For each sample, the best

Sequenza solution was chosen after visual inspection of both the best-fit solution (with the maximum log posterior probability) and alternative solutions.

**Validation of metastatic truncal and non-truncal SNVs from the index whole-genome sequencing case**. Validation was performed using custom pull-down and sequencing of the key mutations identified across the 13 metastases from WGS analysis. The validation experiment was enriched to cover all 2247 metastatic non-truncal variant positions, 652 manually selected metastatic truncal variant positions (identified as either cancer driver mutations or with loss-of-function mutations from the truncal cluster). A 340kbp custom capture probe was designed using Agilent Technologies' online software Sure Select Design Wizard. The highest-stringency repeat masking was used (where possible), as well as a tiling density of 2× and maximum performance boosting (replicating any orphan or GC-rich baits by a higher factor). Agilent ELID ID: 3184291. DNA capture (paired-end, average DNA fragment size 158 bp) libraries were created using native DNA, testing DNA from all 13 whole-genome sequenced metastatic tumours. Libraries were multiplex sequenced to a median depth of 40× on the Illumina MiSeq platform. A variant called in the WGS experiment that was also present in the validation study and supported by at least 2 alternate bases in the validation, is reported as validated somatic.

With these criteria, 7429/7502 (99%) of the metastatic truncal substitutions and 6223/6750 (92%) of the metastatic non-truncal substitutions were validated somatic (the denominator represents the sum of all the SNVs called across all of the 13 samples in the WGS data, excluding those SNVs where coverage in the validation experiment was <30×, see 'Data availability' section). Only 7/7502 of the metastatic truncal mutation calls made in the WGS were not detected in the validation experiment, and 104/6750 (1.5%) of the metastatic non-truncal mutation calls were not detected in the validation experiment. Subsetting only to the metastatic non-truncal substitutions included in the six mutation clusters which passed QC and on which the phylogenetic tree was based (Fig. 2c, d), we found 2127/2231 (95%) SNVs across all 13 metastases were validated (as above, the denominator represents the sum of all the SNVs called across all of the 13 metastases in the WGS data, excluding those SNVs where coverage in the validation experiment was <30×). The breakdown of the validation rate per cluster; Cluster A: 158/192 (82%), Cluster B: 820/837 (98%), Cluster C: 21/24 (88%), Cluster D: 694/720 (96%), Cluster E: 261/276 (95%), Cluster F: 173/182 (95%).

**Targeted sequencing of the archival primary**. The same baits and custom pull-down experiment described above were used to sequence the validation set SNVs within the index autopsy case's primary tumour. DNA was extracted from the two tumour blocks from the same chest wall primary and variants supported by at least 2 alternate bases were called in the primary (samples PD38258u and PD38258v).

**Gene expression analyses**. RNA expression of metastatic tumours from the index autopsy case was determined using the human Affymetrix Clariom D Pico assay (see 'Data and software availability' section). Arrays were analysed using the SST-RMA algorithm on the Affymetrix Expression Console Software. Expression was determined using the Affymetrix Transcriptome Analysis Console. Median absolute deviation normalisation (of probe-level data) was implemented. The Tissue-specific Gene Expression and Regulation (TiGER) database was used to filter out 600 tissue-specific genes[55]. Differential expression was performed using the R package limma (v3.36.1)[56]. Preranked GSEA (GSEA-P) was implemented using the GenePattern module *GSEAPreranked* (v6.0.10)[57]. Hallmark gene sets were downloaded from the MSigDB database[58]. Rank metric was calculated as the sign of log2-FCs calculated using the limma pipeline. The pipeline calculates an enrichment score (ES) that reflects the degree to which a gene set is overrepresented at the extremes (top or bottom) of the entire ranked list. The score is calculated by walking down the list, increasing a running-sum statistic when a gene in the set is encountered and decreasing it when a gene not in the set is encountered. The enrichment score is the maximum deviation from zero encountered in the walk and corresponds to a weighted Kolmogorov–Smirnov-like statistic[57]. The significance of an observed enrichment score (ES) is assessed by comparing the enrichment score with a set of scores computed with randomly assigned phenotypes, which generates a histogram of the corresponding null enrichment scores. The nominal $P$ value is then calculated by using the positive (or negative) portion of the distribution corresponding to the sign of the observed enrichment score. To calculate the false discovery rate, the ES is normalised to account for the size of the gene set yielding a normalised enrichment score. The proportion of false positives is then calculated using the false discovery rate (FDR)[59] corresponding to each normalised enrichment score (NES). The FDR is the estimated probability that a set with a given NES represents a false positive finding; it is computed by comparing the tails of the observed and null distributions for the NES[57]. In order to mitigate the influence of cellular contamination from the surrounding stromal cells in identifying particular genes and biological processes uniquely associated with brain metastases in this patient, we intersected the genes (FDR-adjusted $P$-value < 0.005 and −1<log fold-change>1) and pathways (FDR-adjusted $p$-value < 0.01) significantly differentially expressed between both brain metastases ($n = 5$) and normal tissue (normal samples extracted from the brain and lung, $n = 2$) (Supplementary Fig. 7C), and also between the brain ($n = 5$) and lung metastases

($n = 4$) (Supplementary Fig. 7D). Single sample gene set enrichment analysis (ssGSEA) was employed using the GSVA R package (v1.32.0) to determine the relative enrichment of each of the HALLMARK pathways across samples[60].

**Immune cell deconvolution.** A consensus approach, Consensus[TME] was used to generate cell-type specific estimates of immune cell infiltration from bulk tumour RNA gene expression profiles. This leverages information from multiple gene sets and immune cell expression matrices to build a compendium of robust gene sets for each immune cell type[36]. These genes were further filtered to ensure each has a negative correlation with tumour purity within The Cancer Genome Atlas. Gene sets specific for human skin cutaneous melanoma (SKCM) were used. Finally, single sample gene set enrichment analysis was applied to our Consensus[TME] gene sets to generate normalised enrichment scores for each cell type in each sample. This method has previously been thoroughly benchmarked in a pan-cancer setting[36].

**Extraction of mutational signatures.** SigProfilerMatrixGenerator python packages[61] was used to extract mutational signatures, generating 96 possible mutation types and used to plot mutational profiles. The sigproSS python package (v0.0.0.26)[62] was used to determine the proportion of mutations in each sample attributable to specific COSMIC signatures identified by Alexandrov et al.[31]. sigproSS was also run on the non-truncal mutation clusters defined in the phylogenetic tree (Supplementary Fig. 5).

**Computerised tomography analysis of tumour volume.** Regions of interest were outlined over the entire area of visible tumours on post-contrast CT scans (2 mm thin sections) by a radiologist (FS) using the OsiriX medical imaging software (Pixmeo SARL, Switzerland). Tumour volume was calculated by multiplying the area of tumour outlined on each CT image by the slice thickness.

**Immunohistochemistry.** IHC staining was performed on 5 μm FFPE sections, extracted from the same frozen sections from which DNA and RNA were extracted. Slides were deparaffinised in series of xylene and hydrated in a series of descending ethanol. Heat-induced antigen retrieval was performed using TRIS-EDTA (pH = 9), followed by immunostaining performed on the Leica Bond III autostainer (Leica Biosystems). Antibodies used included mouse monoclonal anti-human CD3 (DAKO, clone F7.2.38, dilution 1:50) and mouse monoclonal anti-human CD8 (DAKO, clone C8/144B, dilution 1:25) at 1 h RT. DAKO REAL[TME] alkaline phosphatase and chromogen red detection system was used for secondary detection of positive staining. Stained slides were counter-stained with haematoxylin and cover-slipped for review. Image acquisition was performed on the Hamamatsu whole slide scanner at 40-fold magnification.

**Statistical analysis and informatics approaches.** All statistical analysis and graphics were generated using R version 3.0.1 (R Foundation for Statistical Computing, Vienna, Austria. URL http://www.R-project.org/). Alignment viewing was performed using Jbrowse and IGV.

**Analysis of Intra-patient tumour heterogeneity (ITH) and phylogenetic tree reconstruction.** To model the clonal structure across all multi-site tumour samples per patient (at WGS, WES and targeted sequencing levels), we used a previously described computational framework[63]. This approach is an SNV-centric ITH analysis which is described below. In the first step, CCF is estimated for each SNV. By taking into account VAF, CNA status of the SNV locus and purity of the tumour sample under analysis, mutation copy number[63], which is the product of CCF and number of SNV-bearing chromosomal segments, was calculated. CCF is then estimated from mutation copy number by adjusting for the number of SNV-bearing chromosomes, as assessed by a binomial distribution maximum likelihood test[63]. The CCF represents the fraction of tumour cells carrying a mutation, and accounts for differences in tumour purity and copy number[4]. SNVs were removed from further analysis if loss of heterozygosity or any other altered CNA status could explain the complete loss of SNV or its differential VAF in other samples. This filtering is essential to eliminate pseudo-heterogeneity being called among the multiple related samples. The second step is to cluster SNVs based on their CCF by using the Bayesian Dirichlet process-based clustering in a multidimensional mode (ndDPClust (https://github.com/Wedge-Oxford/dpclust)[4]) implemented based on DPClust v2.2.8 (https://github.com/Wedge-Oxford/dpclust) to identify clonal and subclonal clusters across multiple samples of the same patient. Other algorithms including that developed by El-Kebir et al.[64] could also be used to infer the evolutionary history of multiple metastatic tumours (see Alves et al.[15]). However, this requires equivalent data from the matched primary tumour, which was not feasible in this case. The DP clusters (identified as local peaks in the posterior mutation density) are then defined as clonal and subclonal according to their CCF peaks (with an expectation of one cluster at CCF of 1 representing clonal variants). Within individual samples, SNVs are annotated as clonal if they are assigned to the cluster with CCF of 1 and subclonal if assigned to a cluster with lower CCF. SNVs are annotated as truncal when they are clonal across all samples from a patient.

The third step is to reconstruct patient-level phylogenetic trees based on all samples. To determine the most likely phylogenetic tree solution, we applied a previously described mathematical framework[4,10]. Specifically, we applied the previously reported sum and crossing rules[65]. Briefly, the sum rule operates upon the premise that if the CCFs of 2 mutation clusters in any sample add up to more than the CCF of their shared ancestral cluster, they must be collinear. The crossing rule states that if 2 mutation clusters B and C are descendants of mutation cluster A, and if cluster B has higher CCF than cluster C in one sample and cluster C has higher CCF than cluster B in another sample, clusters B and C must be branching. Any mutation cluster that violates these two principles is likely to be an artefact and thus removed from tree reconstruction. It should be noted that the sum rule and crossing rule only strictly apply when the infinite sites assumption is assumed. The model states that each mutation only occurs once during the lifetime of a tumour and never reverts to normal[6].

Given that all metastatic samples were clonally related, only one phylogenetic tree was constructed for each patient. Individual sample trees are subtrees of the overall phylogenetic tree, which include just those clones observed within a single sample. One of the strengths of multi-region sampling is that it exerts a greater inferential restriction on possible phylogenies, since the above stated principles must be simultaneously obeyed across all samples from a patient. We reconstructed the phylogenetic trees for all WGS-based and WES-based patients using clusters representing at least 1% of the clustered SNVs. In tree visualisation, the relative branch lengths were made proportional to the fraction of all SNVs assigned to a cluster and the width of each branch was made proportional to the mean CCF of that cluster across all samples of the patient.

**Simulations of phylogenetic trees with variable subclonal heterogeneity.** In order to further validate the lineage divergence observed in the index autopsy case, six simulations were undertaken. Briefly, we simulated trees with a trunk of 100,000 SNVs along with a variable set of non-truncal SNVs across four metastases. We used the same genome-wide coverage distribution (Poisson distribution; lambda = 34) and a similar range of purity as those observed in our WGS dataset for the four tumours (0.7–0.95). Sequencing coverage at each locus was sampled from the Poisson distribution and VAF was simulated by sampling the number of mutant reads from a binomial distribution based on the simulated coverage and success rate adjusted by purity and ploidy to give the desired CCF distributions (clonal and subclonal). Three bifurcations were simulated with equal SNV burden on each branch with the first bifurcation represented by two mutually exclusive clonal clusters (see Supplementary Fig. 8). In addition, two second-step branches were assigned as subclonal with different mean CCFs (0.7 vs. 0.3). The first simulation involved only the truncal cluster with no branching clusters (total SNV = 100,000) while the other five simulations included all the other six clusters but with varying SNV burden on each branch, ranging from low to high SNV burden (i.e., 50, 100, 150, 200, and 500 SNVs). This resulted in the total SNV count to range from 100,300 to 103,000 from simulation two to six.

ndDPClust was not only able to detect all clonal and subclonal clusters at all SNV burden levels (including, for instance, distinguishing 50 variants with mean CCF of 0.7 unique to one sample from 50 clonal variants shared between two samples), but it also did not assign any of the non-truncal variants to the truncal cluster. Moreover, in the truncal-only simulation, ndDPClust did not call any non-truncal clusters from the noise incorporated in generating the truncal cluster (especially at this relatively low WGS coverage). This analysis demonstrates that the non-truncal signals detected in our WGS dataset are detectable by this method and unlikely to be a result of noise in the data.

**Analysis of intra-patient tumour heterogeneity in the primary tumour of the index autopsy case.** As we only had targeted sequencing data on the two FFPE-based primary tumours, and not WGS, we could not confidently call CNAs in these samples. To estimate CCF, we used the protein altering truncal SNVs ($N = 652$) restricted to those that were in regions of the genome that were diploid in all metastatic samples ($N = 144$, 22.2%; closely matching the global proportion of SNVs in diploid regions in all metastatic samples i.e., 24.6%) and diploidy was assumed in the primary samples for those loci. The density distribution of VAF was obtained for both primary samples and the peak with the highest VAF was inferred as the clonal set of variants. Purity was then estimated as twice the VAF of the clonal peak. With inferred CNA status and estimated purity, CCF for each SNV was estimated and ndDPClust was run on both primary samples to detect ITH.

**Driver mutation analyses.** Melanoma drivers were identified as the 20 defined genes with relevant biological evidence outlined in a recently published seminal study[2]. Based on ndDPClust results, driver mutations present in the truncal cluster were assigned as truncal and those present in a non-MRCA cluster which formed a branch were assigned to that branch.

**Reporting summary.** Further information on research design is available in the Nature Research Reporting Summary linked to this article.

## Data availability
The targeted, whole genome and Affymetrix raw sequencing data have been deposited at the European Genome-Phenome Archive (EGA) (https://www.ega-archive.org at the

European Bioinformatics Institute). Data on all somatic SNVs, indels, inversions and copy number calls for both the index WGS autopsy case and the multi-site WES cases have also been deposited at the EGA under the following accession ID's: EGAS00001001348 [ega-archive.org]. Index_autopsy_case: Tumour/normal BAM files for all WGS data. EGAD00001005072. Index_autopsy_case: Caveman, pindel, battenberg and brass calls. EGAD00001005483. Index_autopsy_case: Affymetrix gene expression data. EGAD00010001717. MultiSite_WES_Patients1-7: Tumour/normal BAM files for all WES data. EGAD00001005421. MultiSite_WES_Patients1-7: Caveman, pindel and sequenza calls for all WES data. EGAD00001005487. EGA Study ID: EGAS00001003531 [ega-archive.org]. Index_autopsy_case: Targeted pulldown validation in support of the WGS analysis and targeted pulldown of the primary tumour. EGAD00001005073. The remaining data are available in the Article, Supplementary Information or available from the authors upon request.

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

## Acknowledgements

This work was supported by Cancer Research UK and the Wellcome Trust. The MelResist study is supported by Addenbrooke's Charitable Trust. D.C.W. is supported by the Li Ka Shing foundation and the NIHR Oxford Biomedical Research Centre. The Affymetrix transcriptome sequencing was funded by the Addenbrooke's Charitable Trust pump priming grant ID SWAI/138. We would like to thank the MelResist team for facilitating this research and providing the ethical framework for post-mortem donations, including Emily Barker, Doreen Milne, Catherine Wilson and Myfanwy Nicholas. The Cambridge Brain Bank for facilitating all aspects of the autopsy, particularly Jenny Wilson. The Tissue Bank team at Addenbrooke's Hospital for helping to extract, store and deliver the samples including Martin Bromwich, Emily Daniels, Elizabeth Cromwell and Beverley Haynes. We thank Yvette Hooks for sectioning the primary tumour and digitally scanning the H&E images. The Cancer, Ageing and Somatic Mutation Programme team at the Sanger Institute for facilitating QC and DNA sequencing, particularly Claire Hardy, Stephen Gamble and Elizabeth Anderson. Computation used the Oxford Biomedical Research Computing (BMRC) facility, a joint development between the Wellcome Centre for Human Genetics and the Big Data Institute supported by Health Data Research UK and the NIHR Oxford Biomedical Research Centre. Sam Behjati for his help in searching for balanced inversions (signature of ionising radiation). The views expressed are those of the authors and not necessarily those of the NHS, the NIHR or the Department of Health. Finally, we would like to sincerely thank the patients involved in this study and particularly the index patient and their family. Their complete dedication to advancing melanoma research despite an unpredictable and aggressive disease course dominated every clinical encounter. Their altruism and dedication to help has inspired every aspect of this work and lays the foundation for further studies of this kind.

## Author contributions

R.R. collected the clinical and molecular data, analysed the data and wrote the paper; N.A.P. analysed the DNA sequencing data, undertook the phylogenetic analyses and co-wrote the paper; O.C. helped analyse the Affymetrix expression data and performed the Consensus$^{TME}$ clustering. D.L., F.S. and F.A.G. analysed the imaging data. D.L. also helped format the immune cell IHC images in Fig. 5. M.T. performed the histopathological analyses for the WES multi-site metastases patients. L.M. performed the laser-capture microdissection for the index patients' primary tumour. I.F. helped select H&E images for Fig. 1. K.W. analysed the copy number profiles for WES multi-site cases with sequenza. J.M.M.G. and M.L. performed the immune cell IHC on the autopsy index patient's samples. L.R. ran the mutational signature analyses. M.M. and A.J.S. provided supervision on the expression analyses and Consensus$^{TME}$ clustering. K.A. performed the autopsy and histopathological assessments of the index patients' samples. L.K. helped review clinical aspects of the manuscript. P.C., C.P. and S.J.W. provided overall clinical supervision including in study set up, patient recruitment, as well as critical review of the manuscript. D.C.W. provided overall study supervision, particularly relating to the phylogenetic analyses and contributed to all sections of the manuscript. D.J.A. provided overall study supervision and critical review of the manuscript. All authors approved the final version.

## Competing interests

The authors declare no competing interests.
