## [Peer Review File · Nature Communications]

Reviewers' comments:

Reviewer #1 (Remarks to the Author): Expertise in evolution and phylogenetics

This paper focuses on a WGS multiregional analysis (13+2 samples) of a single melanoma patient. The main result seems to be the demonstration of metastatic "branching evolution" in melanoma. In general, the paper follows a more or less standard approach in cancer genomics, albeit focusing on the mutational clusters and not on the samples, which is positive. However, for the purpose of reconstructing the spread of the tumor, I would have rather used established methods from evolutionary biology, which I believe offers both theoretical and practical advantages, including sound evolutionary models, biological realism, confidence assessment, and reproducibility. Nevertheless, it is up to the authors to consider whether they want to try some of these methods - in my opinion clearly preferable-, or otherwise tone down a bit some of the conclusions, as my impression is that do not always seem to be supported by explicit analyses.

** Major specific comments

1) Being an evolutionary biologist, I cannot ignore the pervasive use of the term "Branching evolution". Evolution is always branched, and therefore "branching" becomes an unnecessary "epitheton ornans". I am aware that this term is pervasive in cancer genomics -in my opinion, due to an unfortunate founder event-, that is often focused only on lineages defined by driver mutations, and that aims to highlight the existence of phylogenetic structure vs. rapid lineage turnover. The latter is often called in the field "linear evolution", which in itself is an oxymoron: lineage replacement can be fast but is never instantaneous. Indeed, sampling bias is very large in cancer genomics, and particularly when analyzing a few genomic targets, it can give the spurious impression that new lineages replace the existing one in a linear fashion. However, the terminology "branching vs linear" is, in my opinion, both misleading and avoidable. I am not aware of any other application of evolutionary biology that uses these terms nowadays. Very few, if any, multiregional WES/WGS cancer studies, describe "linear evolution", which suggests that it is often the result of limited sampling and/or lack of resolution. Certainly, large differences in the level of population structure and rate of lineage turnover exist among tumors. Clearly, the tissue of origin and the strength of selection matter a lot. But in my opinion, this does not justify the use of such terminology. Proper terms exist in evolutionary biology to describe these different scenarios, like "presence/absence of genetic/population/phylogenetic/spatial/temporal structure", "fast/slow (driver) lineage replacement/turnover/extinction", "diversification", "continuous hard/soft selective sweeps", etc. See for example Grenfell et al. Science 2004 review on phylodynamics.

2) Mutational clusters and clonal deconvolution

The authors use an algorithm called ndDPClust to identify clusters of mutations with similar CCF, using all mutations across samples at once. There are a plethora of algorithms out there for the identification of mutational clusters and for clonal deconvolution, and some do take advantage of the information contained in multiple samples, something that ndDPClust does not seem to do. Also, clusters of mutations are not exactly the same as clones, and mutational cluster trees are not necessarily the same as clonal phylogenies. For example, one can have two clones at the same frequency, so the cluster tree will tend to be less resolved than the clonal phylogeny (and in fact tend to suggest more often "linear evolution"). How does ndDPClust, compare with methods that have been developed specifically for clonal reconstruction from multiregional data, like ClonalFinder and Lichee, which outperform other methods in simulations? (see <https://academic.oup.com/bioinformatics/article/34/23/4017/5040314>). It would be nice to show that the main results are not dependent on the particular clustering method used.

2) Phylogenetic reconstruction

Apart from the comments above on cluster trees, the description of the phylogenetic reconstruction is a bit too general. Which software was used to build the phylogenetic trees? If no

software was involved, how can a reader reproduce this analysis? Are then all alternatives considered? Note that the number of solutions for cluster trees can be huge, there can be ties, and confidence in the result is not assessed under this particular methodology.

3) Time

The authors are not really measuring time, so giving that the rate of evolution can change among and within lineages during tumor progression, "early" or "late" cannot really be assessed, we can only say "before" and "after" in some instances. Several temporal statements in the Discussion do not seem to be backed up by specific analyses (e.g. "rapid branching", "short latency"). Time-measured phylogenies can be reconstructed using a different type of methodology.

4) Biogeographic inference

I see a number of statements about geographical spread that in my opinion do not seem supported by the analyses, at least as written. That a large cluster of mutations was shared by all metastatic samples does not immediately mean that a single clone initiated metastatic outgrowth. Multiple, related but distinct metastatic clones could share mutations in this cluster. The authors suggest that the metastatic spread occurred in parallel rather than in a serial fashion, but I did not appreciate any result supporting this statement. The analysis of the primary tumor samples focuses on a subset (~20%) of the SNVs detected in diploid regions across all metastases. The inferred clonal relationships among primary and metastases might be then more complex than those suggested.

The authors infer cluster trees for each sample in order to "infer" the metastatic spread events (sup. figs 2,4). However, this analysis looks subjective. For example, the red arrows in sup. fig 2, which according to the authors represent "polyclonal seeding between samples", seem to have been drawn arbitrarily. Why o-k, and not o-j or o-a? The reconstructed graph in sup. fig. 2 for the left hilar lymph node (o) shows D as the ancestor of B. However, in fig 2D we see that the ancestor of B is A. These graphs are incompatible, and at least one of them has to be wrong.

In my opinion, this set of analyses and conclusions are difficult to justify, particularly when alternative, objective approaches are available. Quantitative, model-based, explicit methods do exist in evolutionary biology to reconstruct biogeographic patterns that can be applied to this type of data. See for example <https://www.nature.com/articles/s41467-019-12926-8>.

5) Evolution of gene expression

Optionally, it would be interesting to implement a phylogenetic analysis of gene expression (see for example <https://www.ncbi.nlm.nih.gov/pmc/articles/PMC3796711/>; <https://www.pnas.org/content/115/3/E409>) in order to detect potentially adaptive parallelisms and convergences, and therefore taking into account the correlations induced by a common history.

6) Role of the microenvironment

In fact, a similar comparative analysis could be implemented to better understand the relationship of the immune compartment on tumor evolution, although this should not be mandatory.

** Minor specific comments

p. 2: that the analyses of multiple biopsies from the same patient results in a more detailed picture of the genetic variation than the analysis of a single biopsy is the inevitable outcome. Isn't this a bit too obvious?

p. 3: the expression "constructing true phylogenies" is confusing. We do not really know whether the inferred phylogenetic tree matches the true phylogeny. I guess they mean "proper" or "legit".

p. 3: "linearly related" cannot be. There is always a branching pattern, regardless of whether we can see it or not.

p. 3: the term "branched lineages" is odd. all lineages are branched, you might mean "divergent".

p. 3: the authors define truncal mutations as being in "all samples from the same patient" but hereafter, truncal mutations seem to be those exclusive of all metastatic samples.

p. 11: "leaves" and "terminal branches" are not synonyms. Terminal branches lead to the terminal nodes or leaves.

p. 19. Which "longitudinal analyses"? Aren't all the metastatic samples contemporaneous?

p. 19. I am not aware of any specific analysis here to detect "clonal selection".

p.19. I would say that truncal variants occur early in tumor evolution by definition, or do you mean "metastatic truncal variants"? Still, the authors do not really know whether these variants occurred "early".

p. 21: lung trunks => long trunks?

p. 29: the second 3) should be 4)

You are welcome to contact me if any clarification is needed.

David Posada (dposada@uvigo.es)

Reviewer #2 (Remarks to the Author): Expertise in melanoma genomics

The manuscript by Rabbie, Ansari-Pour, and colleagues delineates the genetic evolution of metastatic melanoma by sequencing the genomes of multiple macro-metastases and a patient-matched primary from a single patient with melanoma. They also sequenced the exomes of multiple metastases from an extended cohort of patients with melanoma. There are other papers that sequence multiple metastases or metastases and matched primaries from patients with melanoma; however, this study raises the bar over those studies and would be of significant interest to the melanoma research community. As an example, many evolutionary studies do not adequately consider the challenges posed by stromal cell contamination, whereas this was explicitly taken into consideration by the authors of this study.

One of the main conclusions of this study is that metastases evolve in a branched fashion. This was difficult to appreciate from previous studies because the trunks of phylogenetic trees are flooded with UV-radiation-induced mutations, but the authors of this study were able to unequivocally detect this mode of evolution by performing genome sequencing in their index case. I have one concern that may alter this conclusion: did the authors account for copy number deletions? If two lesions share mutations, and then one lesion acquires a deletion, it will appear as if the lesion without the copy number alteration acquired a great number of new mutations, when in fact those "private" mutations were probably part of the trunk of the tree.

Other minor points:

1. I was a bit confused by Figure 2D -- are c and f mislabeled in figure 2D?
2. The following statement is not entirely accurate: "and that the non-truncal metastatic clones

arose de novo during metastatic dissemination” – An alternative possibility could be that there is an even rarer subclone in the primary that was not detectable yet gave rise to the metastases. Considering that the authors only performed 40X sequencing of primary, this is entirely plausible.

3. I disagree with the conclusion from this statement: “Gene expression profiles were also different between tumour and normal tissue from the same organ, indicating that these expression patterns most likely represent changes in tumour cell expression rather than organ site related differences, suggesting an impact of the microenvironment on tumour gene expression” – It is pretty clear from the principal component analyses that the fraction and types of normal cells in a given sample are the dominant variables dictating gene expression. Tumor-cell expression probably plays a smaller role.

4. The following statement was not entirely accurate: “It is therefore our impression that previous interpretations proposing linear evolution of melanoma metastases ... 13,23,39) may have been confounded either by the use of VAF as a surrogate for CCF, or by the lack of power to separate subclones through single sample analyses” – Citation 13 does not propose a linear model metastatic evolution and should not be included as an example of such a study. Citation 13 was not able to comment on linear versus branched modes of evolution because they only evaluated a single metastasis per patient, as Rabbie/Ansari-Pour acknowledge in the second half of their sentence.

Overall, this is an excellent paper that sheds light on the evolution of metastatic melanoma. In addition to the points described above, I complement the authors on finding an interesting index case that was treatment-naïve – the circumstances are unfortunate, but nowadays, it is rare to find a treatment-naïve patient. I also credit the authors on their conclusion that most of the pathogenic mutations occur in the trunks of phylogenetic trees. This is an important point that has been made before; however, there exist studies claiming the opposite (e.g. PMID: 29426936). I encourage the editors to publish these high-quality findings by Rabbie, Ansari-Pour, and colleagues to help settle this debate.

Hunter Shain

Reviewer #3 (Remarks to the Author): Expertise in cancer evolution and phylogenetics

Overview:

In this work the authors present in depth analysis of the evolutionary history of metastatic melanoma patient from whole-genome sequencing data from 13 metastases. From this analysis the authors claim that these metastases are seeded relatively early in the development of the tumor and, moreover, that the evolution of the tumor follows a branching pattern. The authors supplement this analysis with whole-exome data from 7 other patients, and claim that 6 of these also show a branching evolution. They conclude by postulating that branched evolution is characteristic of all melanoma subtypes.

This work describes an interesting dataset (sequencing from 13 metastases) that I suspect that the larger community, especially those working on methods development, will find useful. What appears to be a key decisions by the authors, that appears to have allowed them to uncovered possibly branching evolution, is the removal of all truncal mutations from their analysis. A more thorough explanation of how it can be concluded that the signals found in this data are not just noise is needed and would greatly strengthen these results. Also, the authors may need to consider multiple trees consistent with the datasets. Furthermore, there are a number of places where conclusions drawn by the authors need clarification. This includes how/when polyclonal seeding has been identified and if the authors are claiming anything novel with regards to the use

of CCF over VAFs.

Major Comments:

1. The claim the branched evolution is characteristic of all melanoma subtypes is a broad claim that needs to be better supported. In particular, the authors need to address or at least acknowledge that removing all truncal variants and doing analysis on the remaining variants has the potential to be strongly impacted by noise in the data. Ideally, I'd like to see some sort of simulation to show that these types of signals can actually be detected in this way, although I realize that may be beyond the scope of this work.

2. A number of studies have shown that multiple phylogenetic trees may be consistent with a single dataset (e.g., Jamal-Hanjani M. et al. , N. Engl. J. Med., 2017). The authors claim that only one phylogenetic tree is constructed, but I would be curious if they considered others trees consistent with the data? Inclusion of this analysis would be helpful.

3. While I appreciate the author's conclusion that the use of CCF is essential rather than just using VAF as a proxy for CCF, this conclusion ought to be clarified. Those methods that use VAF typically only use mutations in diploid regions (e.g., Ancestree, CITUP, PhyloSub, and many others) and many others have developed approaches for correcting VAF to account for CNAs (e.g., Canopy, PhyloWGS, SPRUCE, and many others). It seems that the authors are wanting to claim this observation as a novel contribution of this work, but the field is well aware of this limitation. So, additional information on what the authors are hoping to conclude with this point would be helpful.

4. The author's conclusion of polyclonal seeding, Supplementary Figures 2 and 4, are difficult to follow. This needs to be made more clear how these conclusions are being drawn.

Minor Comments:

1. Sometimes the terminology being used is unclear. While the authors do a good job of initially defining truncal, clonal and subclonal, later use of terms such as non-truncal vs. subclonal can be confusing.

2. For the example on page 5 about the order of mutations and copy number gains, it should be made explicit that the mutation is only contained on 2 out of 3 chromosomes in the instance that the copy containing the mutations was the one duplicated, in the other instance only 1 out of 3 will contain the mutation.

3. Page 5 contains an incomplete sentence, "Indeed, whole-genome duplication and other copy number aberrations..."

4. Figure 4 description - what statistical test was used to determine that Oxidative Phosphorylation was the most significant?

5. In the "whole-exome sequencing of multi-site metastases cases" section, the authors first say that Strelka was used to detect indels and later mention PINDEL, is this a typo or were both used?

6. Several methods have been developed recently to infer the evolutionary history of metastatic tumors. These should at least be cited in the Methods section. This includes, El-Kebir et al, Nature Genetics, 2018 and Reiter, J. G. et al., Nat. Commun., 2017.

Reviewers' comments: Reviewer #1 (Remarks to the Author): Expertise in evolution and phylogenetics

This paper focuses on a WGS multiregional analysis (13+2 samples) of a single melanoma patient. The main result seems to be the demonstration of metastatic “branching evolution” in melanoma. In general, the paper follows a more or less standard approach in cancer genomics, albeit focusing on the mutational clusters and not on the samples, which is positive. However, for the purpose of reconstructing the spread of the tumor, I would have rather used established methods from evolutionary biology, which I believe offers both theoretical and practical advantages, including sound evolutionary models, biological realism, confidence assessment, and reproducibility. Nevertheless, it is up to the authors to consider whether they want to try some of these methods—in my opinion clearly preferable—, or otherwise tone down a bit some of the conclusions, as my impression is that do not always seem to be supported by explicit analyses.

We thank the reviewer for their summary and for taking the time to provide these insightful and relevant comments. We entirely agree that there are established reconstruction approaches in the evolutionary biology literature that could be applied in this case. Our approach is based on similar principles rooted in evolutionary biology and have been independently reproduced across several cancer evolutionary studies [e.g. PMID's; 24429703, 25830880, 26099045, 29662167, 31444325, 31907488]. We are nonetheless immensely grateful to the reviewer for highlighting some of these issues relating to our reconstruction approach. The insights are invaluable and highly relevant, and have provided us with the opportunity to improve our definitions and to highlight our key finding (that of previously unrecognised levels of intra-tumoural heterogeneity in melanoma metastases). We have no doubt that these important amendments have improved the rigour of the study and the clarity of our explanations. For this we are extremely grateful.

** Major specific comments (1) Being an evolutionary biologist, I cannot ignore the pervasive use of the term “Branching evolution”. Evolution is always branched, and therefore “branching” becomes an unnecessary “epitheton ornans”. I am aware that this term is pervasive in cancer genomics—in my opinion, due to an unfortunate founder event—, that is often focused only on lineages defined by driver mutations, and that aims to highlight the existence of phylogenetic structure vs. rapid lineage turnover. The latter is often called in the field “linear evolution”,

which in itself is an oxymoron: lineage replacement can be fast but is never instantaneous. Indeed, sampling bias is very large in cancer genomics, and particularly when analyzing a few genomic targets, it can give the spurious impression that new lineages replace the existing one in a linear fashion. However, the terminology “branching vs linear” is, in my opinion, both misleading and avoidable. I am not aware of any other application of evolutionary biology that uses these terms nowadays. Very few, if any, multiregional WES/WGS cancer studies, describe “linear evolution”, which suggests that it is often the result of limited sampling and/or lack of resolution. Certainly, large differences in the level of population structure and rate of lineage turnover exist among tumors. Clearly, the tissue of origin and the strength of selection matter a lot. But in my opinion, this does not justify the use of such terminology. Proper terms exist in evolutionary biology to describe these different scenarios, like “presence/absence of genetic/population/phylogenetic/spatial/temporal structure”, “fast/slow (driver) lineage replacement/turnover/extinction”, “diversification”, “continuous hard/soft selective sweeps”, etc. See for example Grenfell et al. Science 2004 review on phylodynamics.

We thank the reviewer for highlighting this important point which we agree is of central importance. The key message of our study is that CCF clustering across multiple samples reveals that metastatic melanomas are characterised by previously unrecognised levels of intra-tumoural heterogeneity. We agree that the terms “branched” and “linear” evolution, although as the reviewer alludes are commonly used in the cancer literature, may not be entirely accurate or best suited to describe our key finding. We have therefore amended all references to these terms with more appropriate descriptions of intra-tumoural heterogeneity.

Firstly, in the background section, we provide a definition of heterogeneity in this context (p.3):

‘As neoplastic cells proliferate, some of their daughter cells can acquire mutations that convey selective advantages, allowing them to become precursors for new cell lineages⁵. In the metastatic context, dissemination of cells from multiple lineages may lead to admixture of cell populations within multiple sites, likely with different CCFs at each site. By clustering mutations according to their CCFs across multiple samples simultaneously, it is possible to identify cell populations from the same lineage spread across multiple sites. Further, by comparing these

cell populations based on their CCFs across multiple sites simultaneously, it is possible to derive their ancestral relationship. For example, if one cell population is ancestral to another, its CCF must be greater in at least one sample and greater than or equal to the CCF of the descendant cell population in all other samples. It should be noted that by constructing trees from clusters of mutations we avoid the previously-reported inaccurate inferences arising from the construction of sample trees when samples are an admixture of cells from multiple lineages⁶. Moreover, joint analysis of CCFs across multiple samples enables the identification of complex intermixtures of cell populations spread across multiple samples from a primary tumour as well as complex patterns of tumour cell metastasis⁷⁻¹². We note that, throughout this study, we refer to mutations (and mutation clusters) observed in all tumour cells within a sample as ‘clonal’, those found in a subset of tumour cells as ‘subclonal’ and those found clonally in all samples from the same patient as ‘truncal’. We use the term ‘intra-tumoural heterogeneity’ (ITH) to refer to the observation of variants within a tumour that are non-truncal, including variants that may be clonal within some individual samples.’

Further specific uses of the terms **branching/linear evolution** have also been resolved as follows;

Manuscript section	Page	Previous	Replaced with
Title	1	Multi-site clonality analyses uncovers pervasive subclonal heterogeneity and branching evolution across melanoma metastases	Multi-site clonality analysis uncovers pervasive heterogeneity across melanoma metastases
Abstract	2	Through whole-genome sequencing of 13 melanoma metastases sampled at autopsy from a treatment naïve patient and by leveraging the analytical power of multi-sample analyses, we reveal that metastatic cells may depart the primary tumour very early in the disease course and follow a branched pattern of evolution.	Through whole-genome sequencing of 13 melanoma metastases sampled at autopsy from a treatment naïve patient and by leveraging the analytical power of multi-sample analyses, we reveal that metastatic cells may follow a divergent pattern of evolution.
Abstract	2	Multi-sample analyses from a further 7 patients confirmed that branched evolution was pervasive, representing an important mode of melanoma dissemination.	Multi-sample analyses from a further 7 patients confirmed that divergent evolution was pervasive, representing an important mode of melanoma dissemination.
Abstract	2	Our analyses demonstrate that joint analysis of cancer cell fraction estimates across multiple metastases can uncover previously unrecognised levels of subclonal heterogeneity and highlight the limitations of inferring heterogeneity from a single biopsy.	Our analyses demonstrate that joint analysis of cancer cell fraction estimates across multiple metastases can uncover previously unrecognised levels of intra-tumoural heterogeneity and highlight the limitations of inferring heterogeneity from a single biopsy.
Background	3	In particular, two subclones can be either linearly related to each other, or have a	As neoplastic cells proliferate, some of their daughter cells can acquire mutations that convey a

		common ancestor but develop on opposing branched lineages (herein referred to as branched evolution).	selective advantage allowing them to become precursors for new tumour cell lineages⁵. In the metastatic context, dissemination of cells from multiple lineages may lead to admixture of cell populations within multiple sites, likely with different CCFs at each site. By clustering mutations according to their CCFs across multiple samples simultaneously, it is possible to identify cell populations from the same lineage spread across multiple sites. Further, by comparing these cell populations based on their CCFs across multiple sites simultaneously, it is possible to derive their ancestral relationship. For example, if one cell population is ancestral to another, its CCF must be greater in at least one sample and greater than or equal to the CCF of the descendant cell population in all other samples. It should be noted that by constructing trees from clusters of mutations we avoid the previously-reported inaccurate inferences arising from the construction of sample trees when samples are an admixture of cells from multiple lineages⁶. Moreover, joint analysis of CCFs across multiple samples enables the identification of complex intermixtures of cell populations spread across multiple samples from a primary tumour as well as complex patterns of tumour cell metastasis⁷⁻¹³.
Background	5	A picture has therefore emerged whereby the majority	A picture has therefore emerged whereby the majority of

		of mutations in melanoma metastases are truncal and shared by all progeny. Leading up to the formation of a primary melanoma a stepwise model of linear development has been proposed, which includes selection for particular advantageous molecular alterations (including copy number aberrations), facilitating the sequential transition through successive stages ⁹ .	mutations in melanoma metastases are truncal and shared by all progeny. Leading up to the formation of a primary melanoma, a stepwise model of progression has been proposed, which includes selection for particular advantageous molecular alterations (including copy number aberrations), facilitating the sequential transition through successive stages ^{18,19} .
Background	6	We show that metastases in different organs have distinct clonal lineages and conclude that branched evolution likely predominate in melanoma metastases.	We show that metastases in different organs have distinct clonal lineages and conclude that melanoma metastases harbour previously unrecognised levels of ITH .
Results	11	Reconstructing the phylogenetic tree based on the non-truncal mutation clusters uncovers branched evolution .	Reconstructing the phylogenetic tree based on the metastatic non-truncal mutation clusters uncovers divergent evolution .
Results	14	Multi-site clonality analyses from a further 7 patients revealed evidence of pervasive branched evolution across melanoma metastases	Multi-site clonality analyses from a further 7 patients uncovers pervasive evidence of divergent evolution across melanoma metastases
Results	14	We identified 2-10 distinct clusters per patient with clear evidence of branched evolution across 6/7 patients, evidenced by the presence of mutation clusters in mutually exclusive subsets of samples (Supplementary Fig. 2).	We identified 2-10 distinct clusters per patient with clear evidence of divergent evolution across 6/7 patients, evidenced by the presence of mutation clusters in mutually exclusive subsets of samples (Supplementary Fig. 2). By reconstructing sample-level phylogenetic trees, we identified distinct clonal lineages within each patient and found evidence

			for polyclonal seeding in two patients (Supplementary Fig. 4).
Results	14	Given that branched evolution was detected in 6/7 cases (including the acral melanoma patient) based on WES – which has a much lower genomic resolution than WGS – and with as little as two samples per patient in most cases (Supplementary Table), provides strong support that this mode of clonal evolution is likely to be pervasive in melanoma dissemination.	Given that divergent evolution was detected in 6/7 cases (including the acral melanoma patient) based on WES – which has a much lower genomic resolution than WGS – and with as little as two samples per patient in most cases (Supplementary Table), we provide strong evidence that ITH is likely to be pervasive in melanoma metastases.
Discussion	20	We found evidence of branched evolution across metastatic melanoma exomes from a further 6 out of 7 patients (including in one case of metastases from an acral primary), indicating that, even with a much lower sequencing depth and coverage, and nearly two orders of magnitude fewer SNVs (relative to whole-genome sequencing), detailed clonal lineages could still be inferred, and branching evolution is pervasive.	We found evidence of divergent evolution across metastatic melanoma exomes from a further 6 out of 7 patients (including in one case of metastases from an acral primary), indicating that, even with a much lower sequencing depth, and nearly two orders of magnitude fewer SNVs (relative to whole-genome sequencing), detailed clonal lineages could still be inferred, and subclonal heterogeneity is pervasive.
Discussion	20	The detection of branched evolution using lower-resolution WES from archival formalin-fixed paraffin embedded (FFPE)-derived samples is particularly relevant to clinical practice, where the majority of samples are still stored in paraffin, and where custom pull-down is much more readily available than whole-genome sequencing	The detection of ITH using lower-resolution WES from archival formalin-fixed paraffin embedded (FFPE)-derived samples is particularly relevant to clinical practice, where the majority of samples are still stored in paraffin, and where custom pull-down is much more readily available than whole-genome sequencing approaches ⁴¹ .

		approaches ⁴⁰ .	
Discussion	20	Therefore, we postulate that branched evolution is characteristic of all melanoma subtypes.	Therefore, we postulate that multi-sample analyses may reveal that subclonal heterogeneity is characteristic of other melanoma subtypes, although further studies in these rarer subtypes are warranted.
Discussion	20	It is therefore our impression that previous interpretations proposing linear evolution may have been confounded either by the use of VAF as a surrogate for CCF, or by the lack of power to separate subclones through single sample analyses (rather than by the limits of resolution of targeted sequencing approaches employed by many of these studies).	It is therefore our impression that previous studies suggesting that melanoma metastases lack heterogeneity may have been confounded either by the use of VAF as a surrogate for CCF ²⁵ , or by the lack of power to separate subclones through single sample analyses ⁴⁰ (rather than by the limits of resolution of targeted sequencing approaches).
Discussion	23	In summary, through leveraging the power of clonality analyses across multiple whole-genomes we were able to identify rich clonal architectures and uncover pervasive branched evolution of melanoma metastases obtained at autopsy of a single patient, a structure which would not have been evident through single-site reconstructions. This pattern of phylogenetic branching was also evident in exome sequenced metastases obtained from 6 out of 7 additional melanoma patients, one of which was an acral melanoma, suggesting that this is independent of sequencing coverage, depth,	In summary, through leveraging the power of clonality analyses across multiple whole-genomes we were able to identify rich clonal architectures and uncover ITH of melanoma metastases obtained at autopsy of a single patient, a structure which would not have been evident through single-site reconstructions. Using the same approach, we found further pervasive evidence of divergent evolution in exome sequenced metastases obtained from 6 out of 7 additional melanoma patients, one of which was an acral melanoma, suggesting that this is independent of sequencing coverage or depth. Our ability to detect distinct clonal lineages was greatly enhanced by

		the number of SNVs, or melanoma subtype. Our ability to detect distinct clonal lineages was greatly enhanced by leveraging the power from multiple samples. Our data reframes current models of metastatic dissemination and should serve as a cautionary tale in future phylogenetic analyses that define trunk and branch mutations by the presence or absence of shared variants and that do not consider CCF calculations (integrating information from somatic VAFs with tumour purity and ploidy considerations). Future large-scale studies incorporating clonal analyses across multiple metastases will be required to further delineate how these tumours evolve, and provide insights into whether interrupting this process could contribute to patient management.	leveraging the power from multiple samples and, for the first time, uncovers conclusive evidence of ITH in melanoma metastases. Future large-scale studies incorporating clonal analyses across multiple metastases will be required to further delineate how these tumours evolve, and provide insights into whether interrupting this process could contribute to patient management.
Figure 2 legend	24	Subtracting the clonal cluster of variants revealed clonal diversification and branching evolution in the index autopsy case.	Subtracting the clonal cluster of variants revealed subclonal diversification in the index autopsy case.
Figure 3 legend	26	Multi-dimensional Dirichlet processing across metastases from a further 7 patients uncovers pervasive branched evolution.	Multi-dimensional Dirichlet processing across metastases from a further 7 patients uncovers evidence of divergent evolution.

We sincerely thank the reviewer for this suggestion.

2) Mutational clusters and clonal deconvolution – The authors use an algorithm called ndDPCLust to identify clusters of mutations with similar CCF, using all mutations across samples at once.

There are a plethora of algorithms out there for the identification of mutational clusters and for clonal deconvolution, and some do take advantage of the information contained in multiple samples, something that ndDPCLust does not seem to do. Also, clusters of mutations are not exactly the same as clones, and mutational cluster trees are not necessarily the same as clonal phylogenies. For example, one can have two clones at the same frequency, so the cluster tree will tend to be less resolved than the clonal phylogeny (and in fact tend to suggest more often “linear evolution”). How does ndDPCLust, compare with methods that have been developed specifically for clonal reconstruction from multiregional data, like ClonalFinder and Lichee, which outperform other methods in simulations?

(see <https://academic.oup.com/bioinformatics/article/34/23/4017/5040314> [academic.oup.com]). It would be nice to show that the main results are not dependent on the particular clustering method used.

We thank the reviewer for highlighting this point. While the first implementation of Dirichlet process clustering by members of our team was applied to single samples (PMID 22608083, 2012), all subsequent publications have used an algorithm that explicitly models CCFs across multiple samples simultaneously. Within the recently published Pan Cancer Analysis of Whole Genomes working group on Evolution and Heterogeneity (PCAWG-11) study (PMID 32025013; doi.org/10.1101/312041), DPCLust was compared with 10 other subclonal reconstruction algorithms, including many of the algorithms assessed within the paper referred to by the reviewer. DPCLust was found to be amongst the best performing algorithms when applied to simulated data and one of the algorithms that was closest to the consensus of all 11 algorithms on real data.

In order to assess the ability of ndDPCLust to distinguish truncal and non-truncal variants within our dataset, we undertook a separate validation experiment focussed specifically on the 2247 unique metastatic non-truncal mutated positions represented across the 13 metastases. In doing this we found that 6223/6750 (92%) of the non-truncal SNVs were validated (p.37). While not as high as the validation rate for truncal mutations (7429/7502, 99%), this indicates a low rate of artefacts within the non-truncal SNV set. If we subset these variants to the unique non-truncal SNVs assigned to the six mutation clusters that passed QC and were used to construct the phylogenetic tree (1056/2247; Figs 2C & D), we found a higher validation rate (2127/2231,

95%). Looking at the validation rate per cluster, we can further see that this was equally high across all six non-truncal mutation clusters, giving confidence in these six mutation clusters identified. We have added further details to the methods section describing the validation of truncal and non-truncal SNVs from the WGS analysis (p.37). We also found that only 7/7502 of the metastatic truncal mutation calls made in the WGS were not detected in the validation experiment, and 104/6750 (1.5%) of the metastatic non-truncal mutation calls were not detected in the validation experiment.

In addition, we undertook simulations demonstrating that both the truncal and non-truncal mutation clusters detected in the metastases of the autopsy index case were unlikely to be a result of noise in the data (methods p.43). Briefly, we simulated trees with a trunk of 100,000 SNVs along with a variable set of non-truncal SNVs across 4 related samples. We used the same genome-wide coverage distribution and a similar range of purity as those observed in our WGS dataset to simulate VAF of SNVs and compute CCFs. Three bifurcations were simulated (see Figure A, below) with equal SNV burden on each branch. In addition, two second-step branches were assigned as subclonal with different mean CCFs (0.7 vs 0.3).

Figure A. Simulated phylogenetic tree with truncal, clonal and subclonal clusters across four related samples.

Results of ndDPClust from the first simulation involving the truncal cluster only did not call any non-truncal clusters. The other five simulations of a truncal cluster and six non-truncal clusters with varying SNV burdens (ranging from 50 to 500 SNVs on each branch) showed that ndDPClust is able to detect non-truncal clusters at all SNV burdens and similarly did not assign any of the non-truncal variants to the truncal cluster and vice versa (see Table A below).

Cluster	sampleA	sampleB	sampleC	sampleD
A	100 (0)	100 (0)	100 (0)	100 (0)
B	NP	NP	100 (0)	100 (0)
C	NP	NP	100 (0)	NP
D	NP	NP	NP	100 (0)
E	100 (0)	100 (0)	NP	NP
F	100 (0)	NP	NP	NP
G	NP	100 (0)	NP	NP

Table A. Summary of ndDPClust simulation results on mutation assignment to simulated clusters with varying mutation burden. Values represent the mean percentage of variants assigned correctly to each cluster (across samples that harbour that cluster) and the respective standard deviation in brackets. No variant was mis-assigned to any other cluster. NP: Not present.

The correct non-truncal clones were identified in every simulation at the expected CCFs (see Table B), despite incorporating such small cluster sizes (with as little as 50 SNVs; see Table C) and ndDPClust was able to correctly distinguish clonal from subclonal clusters.

Cluster	sampleA	sampleB	sampleC	sampleD
A	1.004 (0.009)	1.004 (0.008)	1.002 (0.009)	1.003 (0.009)
B	0 (0)	0 (0)	1.000 (0.022)	1.002 (0.013)
C	0 (0)	0 (0)	0.999 (0.007)	0 (0)
D	0 (0)	0 (0)	0 (0)	0.304 (0.008)
E	0.999 (0.038)	0.994 (0.008)	0 (0)	0 (0)
F	1.002 (0.038)	0 (0)	0 (0)	0 (0)
G	0 (0)	0.711 (0.010)	0 (0)	0 (0)

Table B. Summary of ndDPClust simulation results on CCF estimation for simulated clusters with varying mutation burden. Values represent the mean CCF of each cluster (across simulations that included that cluster) and the respective standard deviation in brackets.

Cluster	SIM50sampleA	SIM50sampleB	SIM50sampleC	SIM50sampleD	no.of.muts.assigned
A	0.996	1.001	1.005	1.003	100000
B	0.000	0.000	1.004	1.004	50
C	0.000	0.000	0.993	0.000	50
D	0.000	0.000	0.000	0.306	50

E	0.954	0.996	0.000	0.000	50
F	1.050	0.000	0.000	0.000	50
G	0.000	0.713	0.000	0.000	50

Table C. The ndDPClust results for the simulated phylogenetic tree with each non-truncal cluster having a mutation burden of 50 SNVs. Values in columns 2 to 5 represent the CCF assigned to each cluster in each sample and those in column 6 represent the number of variants assigned to each of the detected clusters.

Taken together, this confirms that the mutation clusters on which the phylogenetic tree was constructed are likely to be real and unlikely to have arisen as a result of noise in the data.

2) Phylogenetic reconstruction—Apart from the comments above on cluster trees, the description of the phylogenetic reconstruction is a bit too general. Which software was used to build the phylogenetic trees? If no software was involved, how can a reader reproduce this analysis? Are then all alternatives considered? Note that the number of solutions for cluster trees can be huge, there can be ties, and confidence in the result is not assessed under this particular methodology.

We thank the reviewer for highlighting this concern which we have fully considered. We entirely agree with these comments and our approach to phylogenetic reconstruction certainly considers alternative tree solutions [PMID's; 24429703, 26099045, 29662167, 31444325, 31907488]. After assigning mutation clusters, we manually apply hierarchical ordering of mutation clusters using the previously reported Sum and Crossing rules (PMID: 24484323) to place clusters on shared or branching lineages. Briefly, the sum rule operates upon the premise that if the CCFs of 2 mutation clusters in any sample add up to more than the CCF of their shared ancestral cluster, they must be collinear. The crossing rule states that if 2 mutation clusters B and C are descendants of mutation cluster A, and if cluster B has higher CCF than cluster C in one sample and cluster C has higher CCF than cluster B in another sample, clusters B and C must be branching. In general, the sum and crossing rules do not restrict the space of possible trees to a single candidate. However, for all of the cases reported within this study, this was the case, primarily because all of our cases possessed 1 mutation cluster that was clonal in

a subset of samples and another that was clonal in a different subset of samples. From the crossing rule, such clusters are necessarily branching. We illustrate our method with one example, the index autopsy patient, which we now describe in more detail (p.11). The CCF distribution plot in figure 2C shows how coloured mutation clusters (depicted in columns) are distributed across the 13 metastases (depicted in rows). The mutation clusters showed clear lineage separation, such that samples harbouring clusters from one lineage were mutually exclusive from samples harbouring clusters in the opposing lineage (Fig. 2C). In particular, cluster B (light green) was present (and fully clonal) in the first 7 samples belonging to the first clonal lineage and absent in the latter 6 samples belonging to the second clonal lineage (which in-turn were represented by cluster D (red)). The tree therefore splits into two divergent branches across these mutually exclusive clusters, with both clusters further separating into two terminal branches leading to terminal nodes (cluster C belonging only to the first clonal lineage, clusters E and F only to the second) (Fig. 2C & D, see Methods for further details). (p.11).

We have also added further detail to the methods section (p.42):

'The third step is to reconstruct patient-level phylogenetic trees based on all samples. To determine the most likely phylogenetic tree solution, we applied a previously described mathematical framework^{4,8}. Specifically, we applied the previously reported 'sum' and 'crossing' rules⁶⁴. Briefly, the sum rule operates upon the premise that if the CCFs of 2 mutation clusters in any sample add up to more than the CCF of their shared ancestral cluster, they must be collinear. The crossing rule states that if 2 mutation clusters B and C are descendants of mutation cluster A, and if cluster B has higher CCF than cluster C in one sample and cluster C has higher CCF than cluster B in another sample, clusters B and C must be branching. Any mutation cluster that violates these two principles is likely to be an artefact and thus removed from tree reconstruction.'

For further clarity, we have depicted the same data as CCF histograms (below). Here again, mutation clusters are represented in columns and samples in rows. The x-axis represents log₁₀ of the CCF, such that values on the right of this axis are fully clonal and those to the left are

subclonal. This again shows that the lineages represented by clusters B and D are present clonally in mutually exclusive sets of samples.

The CCF distribution plot for MultiSite_WES_Patient1 (Supplementary Fig. 3) also revealed the same pattern, whereby clusters B (pink) and F (teal blue) were clonal across separate samples and represented mutually exclusive clonal phylogenies at the first bifurcation of the phylogenetic tree (Figure 3). We have added this explanation in brief in the legend to Supplementary figure 3 for further clarity.

3) Time – The authors are not really measuring time, so giving that the rate of evolution can change among and within lineages during tumor progression, “early” or “late” cannot really be assessed, we can only say “before” and “after” in some instances. Several temporal statements in the Discussion do not seem to be backed up by specific analyses (e.g. “rapid branching”, “short latency”). Time-measured phylogenies can be reconstructed using a different type of methodology.

We thank the reviewer for this important comment, which we agree should be clarified. Our analyses focus on timing in relation to the truncal metastatic clone and so any timing references are relative to this. The terms highlighted above are primarily mentioned in one paragraph of the discussion (p.21) and we have replaced these with more exacting terminology:

Previously:

The phylogenetic trees were characterised by non-truncal SNVs appearing late in the evolutionary course and represented by rapid branching of the phylogenetic tree from a long trunk (‘palm tree’ resemblance) (Fig. 2D). Similarly, driver mutations generally arose before subclonal diversification and were found primarily on the lung trunks of the trees (Fig. 2D & 3). This contrasts with what has traditionally been thought to be a slow iterative process of gradual evolution, as typified by recent reports in prostate cancer⁹, where branching generally occurred earlier and more gradually throughout the tumours’ evolutionary trajectories, as well as driver mutations being frequently observed subclonally (rather than predominantly clonally in this and other analyses^{13,15}).

Replaced with:

‘The phylogenetic trees were dominated by long trunks, with smaller branches representing subclonal diversification (‘palm tree’ resemblance) (Fig. 2D). Driver mutations generally arose before subclonal diversification and were found primarily on the long trunks of the trees (Fig. 2D & 3). This contrasts with recent reports in prostate cancer⁹, where branching generally occurred throughout the tumours’ evolutionary trajectories, and with studies of various other tumour types reporting the frequent occurrence of subclonal driver mutations, but is concordant with previous studies of melanoma^{14,16}.’

Other, more isolated references to timing have also been amended:

Abstract (p.2):

Previously: *'Through whole-genome sequencing of 13 melanoma metastases sampled at autopsy from a treatment naïve patient and by leveraging the analytical power of multi-sample analyses, we reveal that metastatic cells may depart the primary tumour very **early in the disease course** and follow a branched pattern of evolution.'*

Replaced with: *'Through whole-genome sequencing of 13 melanoma metastases sampled at autopsy from a treatment naïve patient and by leveraging the analytical power of multi-sample analyses, we reveal that metastatic cells may follow a divergent pattern of evolution.'*

Background (p.19)

Removed: *'Truncal variants occurred early in tumour evolution and dominated downstream phylogenetic reconstruction analyses.'*

Discussion (p.20)

Removed: *'Although the index patient was initially diagnosed with a low-risk stage IB cutaneous primary, predicted to have >95% 5-year survival⁴⁷, the time from detection of metastatic spread to death from disease was very short, which is not uncommon and contributes to the challenges of managing this disease. Further prospective studies will be required to confirm our findings, suggesting a short latency between emergence of the invasive clone and widespread metastases, and to determine how they interplay with the established melanoma prognostic markers⁴⁷.'*

Discussion (p.21)

Previous: *'A recent detailed multi-regional clonality analysis in uveal melanomas has also found multiple driver mutations in **late** branches of the phylogenetic trees, suggesting that these melanomas also continue to evolve as they progress from primary to metastatic disease³².'*

Replaced with: *'A recent detailed multi-regional clonality analysis in uveal melanomas has also found multiple driver mutations in the branches of the phylogenetic trees, suggesting that these melanomas also continue to evolve as they progress from primary to metastatic disease⁴³. Therefore, we postulate that multi-sample analyses may reveal that ITH is characteristic of other melanoma subtypes, although further studies in these rarer subtypes are warranted.'*

4) Biogeographic inference I see a number of statements about geographical spread that in my opinion do not seem supported by the analyses, at least as written. That a large cluster of mutations was shared by all metastatic samples does not immediately mean that a single clone initiated metastatic outgrowth. Multiple, related but distinct metastatic clones could share mutations in this cluster. The authors suggest that the metastatic spread occurred in parallel rather than in a serial fashion, but I did not appreciate any result supporting this statement. The analysis of the primary tumor samples focuses on a subset (~20%) of the SNVs detected in diploid regions across all metastases. The inferred clonal relationships among primary and metastases might be then more complex than those suggested. The authors infer cluster trees for each sample in order to “infer” the metastatic spread events (sup. figs 2,4). However, this analysis looks subjective. For example, the red arrows in sup. fig 2, which according to the authors represent “polyclonal seeding between samples”, seem to have been drawn arbitrarily. Why o-k, and not o-j or o-a? The reconstructed graph in sup. fig. 2 for the left hilar lymph node (o) shows D as the ancestor of B. However, in fig 2D we see that the ancestor of B is A. These graphs are incompatible, and at least one of them has to be wrong. In my opinion, this set of analyses and conclusions are difficult to justify, particularly when alternative, objective approaches are available. Quantitative, model-based, explicit methods do exist in evolutionary biology to reconstruct biogeographic patterns that can be applied to this type of data. See for example <https://www.nature.com/articles/s41467-019-12926-8>. [nature.com]

We thank the reviewer for raising these important points. We have addressed each in-turn:

‘Single clone initiated metastatic outgrowth’

The focus of this sentence is to emphasise that the dominant truncal cluster of mutations initially limits subclonal clustering by CCF, thereby seemingly implying that ITH was absent, as previously suggested [<https://doi.org/10.1101/312041>]. We agree however that this does not necessarily mean that this is the only clone that initiated metastatic outgrowth and have removed this part of the sentence (p.9).

‘Metastatic spread occurring in a parallel rather than a serial fashion’

This statement was made only in reference to a previous study [PMID 26286987] and not to describe any of our findings. This study used this term to summarise their key finding - that genetically distinct subpopulations in the primary tumours were often shared with some locoregional relapses and not with others, concluding that these metastases most likely arose from independent cells in the primary [PMID 26286987]. This study does have important limitations. Nevertheless, it is important work to cite in this field. We agree, however, that the term “parallel dissemination” could cause confusion and have edited the manuscript accordingly (p.21).

Previous:

‘Analysing skin/subcutaneous metastases in 8 patients with cutaneous melanoma, Sanborn and colleagues previously showed that locoregional relapses arose from different subpopulations of the primary tumour cells, which often disseminated in a parallel rather than serial fashion⁴³.’

Replaced with:

‘Analysing skin/subcutaneous metastases in 8 patients with cutaneous melanoma, Sanborn and colleagues previously showed that locoregional relapses arose from different cellular subpopulations of the primary tumour⁴⁴.’

Phylogenetic analyses of the primary tumour

We understand the reviewer’s concerns regarding the phylogenetic analysis of the primary tumour, which they have correctly pointed out was limited to only 144/652 truncal variants in diploid regions. The reason for this restriction is that the primary tumour was represented by two thin (1.2mm) skin tumour samples embedded in paraffin, from which we could only isolate a small number of cells using laser capture microdissection technologies. As a result, we could not confidently call CNAs, which precluded the primary tumour from the initial multi-sample clustering analyses. Instead we designed a custom capture bait set to interrogate the truncal/non-truncal clones identified across the 13 metastases. Only 1 of the (N=2247) non-truncal cluster variants identified in the metastases was present in the primary at this depth, we therefore focused our clustering analyses on the protein-altering truncal SNVs (N=652). Our clustering analysis was restricted to those SNVs that were diploid in all metastatic samples

(N=144, 22.2%; closely matching the global proportion of SNVs in diploid regions in all metastatic samples i.e. 24.6%) and diploidy was assumed in the primary samples for those loci.

Dirichlet process clustering of these 144 variants assigned 37 of them to a subclonal cluster that was clearly separate from the clonal cluster containing the remaining variants (CCF 0.25 95% CI 0.22-0.37 and 0.29 95% CI 0.18-0.34 for the two primary samples PD38258u and PD38258v respectively), and the same variants were identified as subclonal in both primary samples. Therefore, based on evidence from the truncal SNVs across all 13 metastatic tumours, we are confident that this subset of 37 SNVs are subclonal in the primary tumour. It is also important to point out that analyses of the primary are of particular interest in this clinical context. This patient unfortunately developed widespread multi-organ disease from a primary with histopathology that indicated >95% likelihood of survival at 5 years (PMID: 29028110). This is not uncommon in melanoma, which can sometimes follow an aggressive and unpredictable course despite a lower pathologic stage. Due to the difficulties in sequencing cutaneous tumours (partially outlined above), detailed evolutionary insights in primary melanoma are lacking and are of interest to the community.

We do however agree that we can only comment on what we see and these findings on just a subset of SNVs are not yet conclusive. To this end we have adjusted the interpretation of these results to only highlight the key finding that metastatic truncal clones were found as a subclone in the primary. We have also explicitly stated that these findings are not yet conclusive (p.13).

We have firstly added a subtitle to this section (p.13):

'Metastatic truncal mutations were observed as a subclone in the primary tumour'

We have further adjusted the conclusion to this section (p.13):

Previously:

'Taken together, this indicates that the long trunk of the phylogenetic tree originated from this subclonal cluster within the primary tumour, and that the non-truncal metastatic clones arose de novo during metastatic dissemination (Fig. 2D).'

Replaced with:

'This suggests that the long trunk of the phylogenetic tree could have originated from this subclonal cluster within the primary tumour. However, the small number of evidential variants warrants further studies in primary and metastatic melanomas (Fig. 2D).'

In addition, we have removed the sentence below in the discussion (p.21), which we agree would require further analysis.

Removed 'Our finding that truncal mutations were identified as a subclone within the index patients' cutaneous primary further corroborate the observation that metastases likely seeded early, at a time when distant disease was clinically undetectable.'

We hope that these adjustments will allow these findings to be placed in their appropriate context and that this could help stimulate further research in this area.

Polyclonal seeding and sample level phylogenetic trees

We thank the reviewer for highlighting this important issue relating to better clarity in the explanations of polyclonal seeding, which we have addressed in both in the main text and in the supplementary figures.

The first mention of polyclonal seeding is in the introduction, where we refer to a recent multi-site exome sequencing study reporting that polyclonal seeding in melanoma was rare [PMID 29991680] (p.4). We have used this opportunity to outline a definition of this term, which we hope will better introduce this concept. This sentence now reads (p.4):

'A recent whole-exome sequencing (WES) study of 86 distant metastases obtained from 53 patients used variant allele frequency (VAF, proportion of reads supporting a mutant allele in parallel sequencing data) of shared versus private mutations in each lesion to infer the likely clonal status of private mutations within each sample¹⁷. Although many private mutations were subclonal, this study found polyclonal seeding (defined as a sample harbouring subclonal mutations from 2 or more diverged clonal lineages, thus representing multiple seeding events by two or more genotypically distinct cells¹³) to be a rare event¹⁷.

We next refer to polyclonal seeding in supplementary figures 2 and 4 and have made important additions to both the figures and legends. In supplementary figure 2, we have included the phylogenetic tree at the top of the figure and have colour-coded the clusters enabling easier tracing. Firstly, it is important to point out that these sample-level phylogenetic trees only show the subclones present within each sample and so are just a sample-level representation of the overall phylogenetic tree. The reason for showing this is to highlight how the samples separate based on their distinct clonal lineages. To this end, we have grouped the samples into their respective clonal lineage, with the samples from the first branch of the phylogenetic tree (emanating from cluster B, light green) highlighted as belonging to the first clonal lineage and samples from the second main branch of the phylogenetic tree (emanating from cluster D, red) highlighted as belonging to the second clonal lineage. With this depiction, we believe it is now much clearer to visualise those samples that have evidence for polyclonal seeding. In the legend, we have also more clearly outlined how this relates to the relevant samples (p.28):

‘Supplementary Fig 2. Sample-level phylogenetic tree for the index autopsy case. Each tree represents a subtree of the overall phylogenetic tree (Fig. 2D) including just those subclones seen within that particular sample. However in doing this we were able to segregate the samples based on their respective clonal lineages. We observed two clear lineages, representing distinct waves of metastatic seeding depicted here as the lineage 1 and 2 emanating from clusters B (light green) and clusters D (red) respectively. Seeding events are represented with dashed arrows. When the sequence of clusters seed across samples from subclonal to clonal, unidirectional arrows are used, whereas bidirectional arrows represent seeding in either direction. That cluster F (purple) is subclonal in more than one metastasis across both branched lineages (in sample F from the first branched lineage and in samples C, E and D from the second lineage) suggests that polyclonal seeding has occurred.’

‘Supplementary Fig 4. Sample-level phylogenetic tree for multi-site whole-exome sequenced cases. The respective branched lineages are depicted for each patient. Only two patients (MultiSite_WES_Patient3 and MultiSite_WES_Patient4) displayed polyclonal seeding, evidenced by one sample harbouring subclonal mutation clusters from 2 or more distinct lineages.’

Finally, we have added further detail to the brief reference to polyclonal seeding within the discussion (p.21)

'Interestingly, a single subclonal mutation cluster (cluster F, shown in purple in Supplementary Fig. 2) was found subclonally in both the brain metastases from the index autopsy case, evidencing polyclonal seeding to these sites.'

We hope these additions go some way to better explaining how polyclonal seeding was identified, how we represent it and its relevance to the index patient's story.

Ancestor error in supplementary figure 2 (sample 'O')

We apologise for this inadvertent mistake in depicting sample 'o' within this sample tree. We confirm that the ancestor of B is A and not D. This has now been rectified in Supplementary Figure 2. All relevant arrows from sample 'o' now emanate from cluster B.

5) Evolution of gene expression [2] Optionally, it would be interesting to implement a phylogenetic analysis of gene expression (see for example <https://www.ncbi.nlm.nih.gov/pmc/articles/PMC3796711/>; [\[ncbi.nlm.nih.gov\]](https://www.ncbi.nlm.nih.gov) <https://www.pnas.org/content/115/3/E409> [\[pnas.org\]](https://www.pnas.org)) in order to detect potentially adaptive parallelisms and convergences, and therefore taking into account the correlations induced by a common history. [2][6] Role of the microenvironment [2] In fact, a similar comparative analysis could be implemented to better understand the relationship of the immune compartment on tumor evolution, although this should not be mandatory. [2]

We thank the reviewer for these comments. We agree that gene expression and the role of the microenvironment are important issues in cancer evolution. While they are outside the scope of this manuscript, we intend to explore these data in future studies.

** Minor specific comments [2] p. 2: that the analyses of multiple biopsies from the same patient results in a more detailed picture of the genetic variation than the analysis of a single biopsy is the inevitable outcome. Isn't this a bit too obvious?

We thank the reviewer for this comment which we understand may be an obvious statement, which we have therefore removed (p.3).

[2] p. 3: the expression "constructing true phylogenies" is confusing. We do not really know whether the inferred phylogenetic tree matches the true phylogeny. I guess they mean "proper" or "legit".

We thank the reviewer for highlighting this. This sentence is specifically in reference to our cluster-centric (as opposed to sample-centric) approach. We have therefore amended this statement (p.3):

Previously:

'By comparing the constituent subclonal mutations between pairs of tumours, it is possible to derive the ancestral relationships between subclones rather than between samples, thereby constructing true phylogenies⁶.'

Replaced with:

'In the metastatic context, dissemination of cells from multiple lineages may lead to admixture of cell populations within multiple sites, likely with different CCFs at each site. By clustering mutations according to their CCFs across multiple samples simultaneously, it is possible to identify cell populations from the same lineage spread across multiple sites. Further, by comparing these cell populations based on their CCFs across multiple sites simultaneously, it is possible to derive their ancestral relationship. For example, if one cell population is ancestral to another, its CCF must be greater in at least one sample and greater than or equal to the CCF of the descendant cell population in all other samples. It should be noted that by constructing trees from clusters of mutations we avoid the previously-reported inaccurate inferences arising from the construction of sample trees when samples are an admixture of cells from multiple lineages⁶.'

We hope this provides a clearer description and we thank the reviewer for drawing our attention to this point. ☒

p. 3: "linearly related" cannot be. There is always a branching pattern, regardless of whether we can see it or not. ☒☒p. 3: the term "branched lineages" is odd. all lineages are branched, you might mean "divergent".

Previously:

'This type of quantitative modelling provides much greater resolution than single-sample studies and has yielded important insights into the patterns and timing of tumour cell spread⁷⁻¹¹. In particular, two subclones can be either linearly related to each other, or have a common ancestor but develop on opposing branched lineages (herein referred to as branched evolution).'

Thank you for these two helpful comments, referencing the same sentence in the background (p3). This section has been replaced (see also table and comment above).

p. 3: the authors define truncal mutations as being in "all samples from the same patient" but hereafter, truncal mutations seem to be those exclusive of all metastatic samples.

We thank the reviewer for kindly pointing this out and for highlighting the need for consistency in usage of these key terms (clonal/subclonal/truncal). The confusion has likely arisen as the

initial definition of these terms within the background (p.3) may have been confused with latter uses of these terms specifically referring to the index autopsy case, whereby Dirichlet processing was initially undertaken on the 13 whole-genome sequenced metastatic tumours. As such, we have maintained the initial definitions for these terms within the background section but have clarified these terms in relation to the index autopsy case analyses as follows:

Firstly, in the results section titled ‘Metastatic melanomas are dominated by UV-induced clonal mutations that dominate phylogenetic reconstructions’ (p.9) we have added the following statement:

‘Given that clustering analyses were initially undertaken on the metastatic tumours, subsequent references to metastatic-truncal and metastatic non-truncal mutation clusters apply only to those identified in the metastases of the index autopsy case.’

We further ensured that all subsequent statements in this section referring to truncal/non-truncal mutations specifically refer to the metastases e.g. in the sentence immediately following that above (p.9) we have added:

*‘Using this approach, we found that >90% of all somatic variants in the metastases were metastatic truncal, with only one additional cluster which represented at least 1% of the SNVs (1651, 1.35%). The large metastatic-truncal cluster was dominated by C>T transitions at dipyrimidines (characteristic of UV-induced mutational damage²⁷) and was shared across all metastases, implying that ITH was absent (**Fig. 2A**).’*

As well as the final sentence in this paragraph (p.10):

*We undertook a validation experiment by custom capture pull-down sequencing of non-silent metastatic-truncal SNVs present across all 13 metastases (selected as either cancer driver or loss-of-function SNVs, N=652), as well as all 2247 metastatic non-truncal SNVs. In this way, 99% of metastatic-truncal SNVs and 92% of all the metastatic non-truncal SNVs were observed to be true variants, instilling confidence in the downstream phylogenetic reconstructions based on these SNVs (see **Methods**).*

The subsequent section *'Reconstructing the phylogenetic tree based on the metastatic non-truncal mutation clusters uncovers divergent evolution'* (p.11) also has a number of key amendments clarifying that these analyses pertain to the whole-genome sequenced metastatic tumours. We also hope that the subheading of the analysis of the primary tumour (p.13) *"Metastatic truncal mutations were observed as a subclone in the primary tumour"* further clarifies our use of the term 'metastatic-truncal' in this section.

p. 11: "leaves" and "terminal branches" are not synonyms. Terminal branches lead to the terminal nodes or leaves.

We thank the reviewer for this more accurate terminology which we have amended in this section.

p. 19. Which "longitudinal analyses"? Aren't all the metastatic samples contemporaneous?

We thank the reviewer for this completely correct observation. The reviewer correctly highlights that the phylogenetic analyses and evolutionary tree for the index autopsy case were mainly based upon analyses of the (whole-genome sequenced) metastases sampled at autopsy. We have therefore removed the reference to 'longitudinal' and updated this sentence to read (p.19):

'In our study, analyses of clonal structure from multi-site genome sequenced melanoma metastases provided a powerful method to detect mutation clusters and a unique insight into clonal evolution.'

p. 19. I am not aware of any specific analysis here to detect "clonal selection".

Thank you, we have changed *'...provided a powerful method to detect clonal selection'* to *'provided a powerful method to detect mutation clusters'* as a more accurate reflection of how this multi-site WGS data was used in this context (p.19, full sentence above).

p.19. I would say that truncal variants occur early in tumor evolution by definition, or do you mean “metastatic truncal variants”? Still, the authors do not really know whether these variants occurred “early”.

We agree with this statement. It is important to specify that these are indeed ‘*Metastatic truncal clones*’ as outlined above. We have amended all terminology around timing as per comment (3) above.

p. 21: lung trunks => long trunks? We thank the reviewer for spotting this typo. We have corrected this in the text (p.21):

*‘Similarly, driver mutations generally arose before subclonal diversification and were found primarily on the long **trunks** of the trees (Fig. 2D & 3).’*

p. 29: the second 3) should be 4)

We thank the reviewer for correctly spotting this typo. The reference to the main PCA of protein-coding genes in this analysis should indeed be Fig. 3A. We have corrected this in the text (p.30).

You are welcome to contact me if any clarification is needed. David Posada
(dposada@uvigo.es)

Reviewer #2 (Remarks to the Author): Expertise in melanoma genomics. The manuscript by Rabbie, Ansari-Pour, and colleagues delineates the genetic evolution of metastatic melanoma by sequencing the genomes of multiple macro-metastases and a patient-matched primary from a single patient with melanoma. They also sequenced the exomes of multiple metastases from an extended cohort of patients with melanoma. There are other papers that sequence multiple metastases or metastases and matched primaries from patients with melanoma; however, this study raises the bar over those studies and would be of significant interest to the melanoma research community. As an example, many evolutionary studies do not adequately consider the challenges posed by stromal cell contamination, whereas this was explicitly taken into consideration by the authors of this study.

We thank reviewer 2 for their accurate summary and for their very kind compliments. We consider this reviewer an internationally renowned expert in melanoma genomics and have extensively referenced his studies in this field. Our ability to glean this depth of detail (including accounting for stromal contamination from normal cells) is owed entirely to the patients and their families (all of whom we have treated), who have provided us this unprecedented insight. We are delighted the reviewer remarks that these data would be of significant interest to melanoma community and are excited to share all the raw whole-genome, transcriptome and gene expression data (already deposited at European Genome-Phenome Archive under study ID's EGAS00001001348 & EGAS00001003531, see data and software availability).

The reviewer raises important points of clarification which we have addressed. These have certainly strengthened the descriptions surrounding our approach and the findings.

One of the main conclusions of this study is that metastases evolve in a divergent fashion. This was difficult to appreciate from previous studies because the trunks of phylogenetic trees are flooded with UV-radiation-induced mutations, but the authors of this study were able to unequivocally detect this mode of evolution by performing genome sequencing in their index case. I have one concern that may alter this conclusion: did the authors account for copy number deletions? If two lesions share mutations, and then one lesion acquires a deletion, it will appear as if the lesion without the copy number alteration acquired a great number of new mutations, when in fact those “private” mutations were probably part of the trunk of the tree.

We thank the reviewer for highlighting this issue, which we agree is of central importance and this is certainly accounted for in our analyses. To clarify this approach, we have added the following statement to the analysis of intra-tumoural heterogeneity (p.41):

'In the first step, CCF is estimated for each SNV. By taking into account VAF, CNA status of the SNV locus and purity of the tumour sample under analysis, mutation copy number⁶³, which is the product of CCF and number of SNV-bearing chromosomal segments, was calculated. CCF is then estimated from mutation copy number by adjusting for the number of SNV-bearing chromosomes, as assessed by a binomial distribution maximum likelihood test⁶¹. The CCF represents the fraction of tumour cells carrying a mutation, and accounts for differences in tumour purity and copy number⁴. SNVs were removed from further analysis if loss of heterozygosity or any other altered CNA status could explain the complete loss of SNV or its differential VAF in other samples. This filtering is essential to eliminate pseudo-heterogeneity being called among the multiple related samples.'

Other minor points: 1. I was a bit confused by Figure 2D -- are c and f mislabeled in figure 2D?

The CCF distribution plot in figure 2C shows how coloured mutation clusters (depicted in columns) are distributed across the 13 metastases (depicted in rows). The mutation clusters showed clear lineage separation, such that samples harbouring clusters from one lineage were mutually exclusive from samples harbouring clusters in the opposing lineage (Fig. 2C). In particular, cluster B (light green) was present (and fully clonal) in the first 7 samples belonging to the first clonal lineage and absent in the latter 6 samples belonging to the second clonal lineage (which in-turn were represented by cluster D (red)). The tree therefore split into two divergent branches across these mutually exclusive clusters, with both clusters further separating into two branches leading to terminal nodes (cluster C belonging only to the first clonal lineage, clusters E and F only to the second) (Fig. 2C & D, see Methods for further details). The finding of mutually exclusive clonal clusters at the first divergence means that alternative tree solutions are very unlikely. In order to highlight this in the manuscript, we have added this explanation to the results (p.11).

In order to highlight this to the reader, and in particular to emphasise that the tree drawn in Figure 2D represents the only possible clonal phylogeny, we have added this explanation to this section of the results (see p.11).

Furthermore, to enable easier tracing of the mutation clusters we have further adjusted the relative ordering of the clusters on the tree. In particular, the upper branch of phylogenetic tree now shows the first clonal lineage (represented by clusters B and C), whereas the lower branch of the tree represents the second clonal lineage (represented by clusters D, E and F). We hope this will further facilitate a better appreciation of the cluster ordering in this context.

2. The following statement is not entirely accurate: “and that the non-truncal metastatic clones arose de novo during metastatic dissemination” – An alternative possibility could be that there is an even rarer subclone in the primary that was not detectable yet gave rise to the metastases. Considering that the authors only performed 40X sequencing of primary, this is entirely plausible.

We thank the reviewer for this comment, which we agree is a possibility. We have therefore removed this statement and have also removed the final sentence in this paragraph (p.13).

Previously:

‘Taken together, this indicates that the long trunk of the phylogenetic tree originated from this subclonal cluster within the primary tumour, and that the non-truncal metastatic clones arose de novo during metastatic dissemination (Fig. 2D). These non-truncal metastatic clones that make up the phylogenetic tree therefore represent changes that were acquired in addition to those that establish the primary tumour.’

Replaced with:

‘This suggests that the long trunk of the phylogenetic tree could have originated from this subclonal cluster within the primary tumour. However, the small number of evidential variants warrants further studies in primary and metastatic melanomas (Fig. 2D).’

3. I disagree with the conclusion from this statement: “Gene expression profiles were also different between tumour and normal tissue from the same organ, indicating that these expression patterns most likely represent changes in tumour cell expression rather than organ site related differences, suggesting an impact of the microenvironment on tumour gene expression” – It is pretty clear from the principal component analyses that the fraction and types of normal cells in a given sample are the dominant variables dictating gene expression. Tumor-cell expression probably plays a smaller role.

We thank the reviewer for correctly pointing out this detail. Although the principal component analysis in figure 4D did indicate that there is separation between the tumour samples (circles) and the corresponding patient-matched normal brain/lung samples (triangles) on PC2 along the y-axis, we agree that this explains a relatively smaller variance when compared to the separation of brain metastases (red circles) from lung/liver/cardiac metastases (blue, green and brown circles respectively) on PC1 along the x-axis (12% vs 53% of variance in PC2 vs PC1, respectively). We have therefore deleted this statement from this section of the results (p.17):

Deleted:

‘Gene expression profiles were also different between tumour and normal tissue from the same organ, indicating that these expression patterns most likely represent changes in tumour cell expression rather than organ site related differences, suggesting an impact of the microenvironment on tumour gene expression’

In fact, it is because of this observation that, when presenting the genes/pathways that might be uniquely associated with brain metastases in this patient, we did not just present the result of the differential expression/GSEA of the comparison between the brain vs lung metastases, but we also intersected this result with the same differential expression/GSEA comparing the brain metastases vs normal brain/lung samples. We have therefore acknowledged this issue, by adjusting the next sentence in the **results** section (p.17) to read:

‘In order to identify the tumour-specific genes and biological processes uniquely associated with brain metastases in this patient and mitigate for the potential influence of cellular

contamination from the surrounding stromal cells (Fig. 4A), we intersected the genes (Fig. 4B) and pathways (Fig. 4C) differentially expressed between both brain metastases (n=5) vs normal tissue (from the patients' normal brain and lung tissue) (Supplementary Fig. 7C), with those between brain (n=5) vs lung metastases (n=4) (Supplementary Fig. 7D).'

We have also adjusted the **legend** to figure 4 (p.26), by removing this statement:

'...which in-turn separated from the corresponding patient-matched normal organ control samples. The separation of tumour and normal samples indicates that the regional separation seen between brain and lung metastases is less likely to be confounded by non-tumoural cells.'

Replaced with:

'A regional separation can be seen between the brain metastases (n=5, red circles) and lung metastases (n=4, blue circles) on PC1 along the x-axis, which accounted for greater variance than the separation between the tumour (n=11) and normal samples (n=2) by PC2 along the y-axis (53% vs 12% respectively), indicating that expression patterns are at least partially influenced by cellular contamination from the surrounding stromal cells.'

Finally, for further clarification, we have also updated the methodology for these analyses to highlight this approach, by adding the following sentence in the 'Gene expression analysis' **methods** section (p.39, penultimate sentence of the section):

'In order to mitigate the influence of cellular contamination from the surrounding stromal cells in identifying particular genes and biological processes uniquely associated with brain metastases in this patient, we intersected the genes (FDR-adjusted p-value<0.005 and -1<log fold-change>1) and pathways (FDR-adjusted p-value<0.01) significantly differentially expressed between both brain metastases (n=5) and normal tissue (normal samples extracted from the brain and lung, n=2) (Supplementary Fig. 7C), and also between the brain (n=5) and lung metastases (n=4) (Supplementary Fig. 7D).'

4. The following statement was not entirely accurate: “It is therefore our impression that previous interpretations proposing linear evolution of melanoma metastases ... 13,23,39) may have been confounded either by the use of VAF as a surrogate for CCF, or by the lack of power to separate subclones through single sample analyses” – Citation 13 does not propose a linear model metastatic evolution and should not be included as an example of such a study. Citation 13 was not able to comment on linear versus branched modes of evolution because they only evaluated a single metastasis per patient, as Rabbie/Ansari-Pour acknowledge in the second half of their sentence.

We thank the reviewer for correctly pointing this out, which has helped us further clarify this section. Firstly, we have removed citation (13) from this statement, which indeed did not comment on the modes of melanoma evolution. The study by Gartner *et al.* analysed two sets of metastases from two patients and found that the vast majority of SNVs and CNAs were shared across the metastases, concluding that ‘metastases were derived from the same parental clone that harbored the majority of the genetic alterations and chromosomal instability’ [PMID: 23006843]. The study by Dentre *et al.*, although primarily based on whole-genome sequencing data from single melanoma metastases, did employ phylogenetic reconstruction analyses to conclude that metastatic melanomas may lack subclonal heterogeneity (as outlined at the beginning of the discussion) (<https://doi.org/10.1101/312041>). For this reason, we have kept the references to these two studies in this statement. Importantly though, we have adjusted this statement to move away from descriptions “proposing linear evolution” to “...previous studies suggesting that melanoma metastases lack heterogeneity” as this is a more accurate depiction of these analyses and helps to contextualise the novel findings in our study. Therefore, we have updated this sentence (p.20):

Previously:

‘It is therefore our impression that previous interpretations proposing linear evolution of melanoma metastases (where it has been thought that genetically distinct cell populations in the primary tumour might metastasise sequentially from one site to the next^{13,23,39}), may have been confounded either by the use of VAF as a surrogate for CCF, or by the lack of power to

separate subclones through single sample analyses (rather than by the limits of resolution of targeted sequencing approaches employed by many of these studies)'.

Updated to:

'It is therefore our impression that previous studies suggesting that melanoma metastases lack heterogeneity may have been confounded either by the use of VAF as a surrogate for CCF⁴¹, or by the lack of power to separate subclones through single sample analyses³⁹ (rather than by the limits of resolution of targeted sequencing approaches).'

In correcting this we came to realise that the reference to “linear development” (in reference to the seminal evolutionary studies by Shain *et al.* [PMID's: 26559571 & 27125352]) could also be more accurately depicted. We have therefore also amended the sentence (referencing these landmark studies, background, p.5).

Previously:

“Leading up to the formation of a primary melanoma a stepwise model of linear development has been proposed”

Replaced with:

“Leading up to the formation of a primary melanoma a stepwise model of progression has been proposed, which includes selection for particular advantageous molecular alterations (including copy number aberrations), facilitating the sequential transition through successive stages^{18,19}.”

We hope the reviewer will agree that this provides a more nuanced description of these landmark observations.

Overall, this is an excellent paper that sheds light on the evolution of metastatic melanoma. In addition to the points described above, I complement the authors on finding an interesting index case that was treatment-naïve – the circumstances are unfortunate, but nowadays, it is rare to find a treatment-naïve patient. I also credit the authors on their conclusion that most of

the pathogenic mutations occur in the trunks of phylogenetic trees. This is an important point that has been made before; however, there exist studies claiming the opposite (e.g. PMID: 29426936). I encourage the editors to publish these high-quality findings by Rabbie, Ansari-Pour, and colleagues to help settle this debate.

Hunter Shain^{1,2} We thank the reviewer for this kind compliment. It was the index patients' expressed wish to donate to this study and for others to learn from their case. We are extremely privileged to have been part of this and to be able to share these data for the wider benefit of the community. The findings of driver mutations occurring (almost exclusively) on the trunks of the phylogenetic trees was unequivocal and is a validation of the original observations made by this reviewer. We are pleased to confirm these findings which have important clinical implications.

^{1,2}

Reviewer #3 (Remarks to the Author): Expertise in cancer evolution and phylogenetics

¶Overview:¶In this work the authors present in depth analysis of the evolutionary history of metastatic melanoma patient from whole-genome sequencing data from 13 metastases. From this analysis the authors claim that these metastases are seeded relatively early in the development of the tumor and, moreover, that the evolution of the tumor follows a branching pattern. The authors supplement this analysis with whole-exome data from 7 other patients, and claim that 6 of these also show a branching evolution. They conclude by postulating that branched evolution is characteristic of all melanoma subtypes. ¶¶This work describes an interesting dataset (sequencing from 13 metastases) that I suspect that the larger community, especially those working on methods development, will find useful. What appears to be a key decisions by the authors, that appears to have allowed them to uncovered possibly branching evolution, is the removal of all truncal mutations from their analysis. A more thorough explanation of how it can be concluded that the signals found in this data are not just noise is needed and would greatly strengthen these results. Also, the authors may need to consider multiple trees consistent with the datasets. Furthermore, there are a number of places where conclusions drawn by the authors need clarification. This includes how/when polyclonal seeding has been identified and if the authors are claiming anything novel with regards to the use of CCF over VAFs.

We thank reviewer 3 for their comments, which represent a thorough and accurate summary of our work. The conclusion of our study is that multi-dimensional clustering based on cancer cell fraction uncovered a previously unrecognised levels of intra-tumoural heterogeneity in metastatic melanoma. We have restructured our concluding paragraph to highlight this novelty, focussing much less on these rarer melanoma subtypes. We entirely agree with the reviewer that these postulations are based on limited evidence and we have now explicitly stated that this statement will require further work (see answers to comment 1 below).

We thank the reviewer for highlighting the value of this dataset which remains incredibly rare. We are extremely indebted to the patients (all of whom we have had the honour to treat) who have given us this unprecedented insight. We are proud to share these analyses as well as the raw data with the community, which was their expressed wish (European Genome-Phenome Archive accessible, via study ID's EGAS00001001348 & EGAS00001003531, summarised in data and software availability).

The reviewer correctly points out that the removal of truncal variants was key in uncovering this structure. We have highlighted below (in response to comment 3) how we validated these non-truncal variants, as well as the further simulations we have undertaken showing that these signals are unlikely to be enriching for noise. Although we certainly consider multiple tree solutions and now fully outline our mathematical approach to this, we were further reassured by the unequivocal divergence of the branched lineages, leaving only one possible clonal phylogeny (comment 3). We agree with the reviewer regarding the need for clarity around the observations of polyclonal seeding which we have also addressed (comment 4).

We thank the reviewer for their insightful comments and valuable suggestions. There is no doubt that these amendments have helped us clarify the descriptions and highlight the novel contributions of this work.

Major Comments: 1. The claim the branched evolution is characteristic of all melanoma subtypes is a broad claim that needs to be better supported. In particular, the authors need to address or at least acknowledge that removing all truncal variants and doing analysis on the remaining variants has the potential to be strongly impacted by noise in the data. Ideally, I'd like to see some sort of simulation to show that these types of signals can actually be detected in this way, although I realize that may be beyond the scope of this work.

We thank the reviewer for this important comment. The postulation of divergent evolution being characteristic of all melanoma subtypes is based primarily on our findings using multi-sample (n=6) analyses on the acral melanoma patient in this study (MultiSite_WES_Patient1), as well as a recent seminal publication on the evolution of uveal melanomas [PMID: 31253977]. We entirely agree however that this is based on limited evidence from these (much rarer) subtypes. The primary aim of this statement however is to highlight the benefit of multi-dimensional clustering in unravelling new clonal structures which we hope could ultimately be applied to other tumours. We have therefore amended this statement (p.20).

Previously:

'Therefore, we postulate that branched evolution is characteristic of all melanoma subtypes.'

Replaced with:

'Therefore, we postulate that multi-sample analyses may reveal that ITH is characteristic of other melanoma subtypes, although further studies in these rarer subtypes are warranted.'

We further refer the reviewer to the answers made to reviewer 1 comment number 2 (*Mutational clusters and clonal deconvolution*), where we highlight our validation of the mutational clusters, as well as further simulation experiments demonstrating that both the truncal and non-truncal mutation clusters detected in the metastases of the autopsy index case were unlikely to be a result of noise in the data.

2. A number of studies have shown that multiple phylogenetic trees may be consistent with a single dataset (e.g., Jamal-Hanjani M. et al. , N. Engl. J. Med., 2017). The authors claim that only one phylogenetic tree is constructed, but I would be curious if they considered others trees consistent with the data? Inclusion of this analysis would be helpful.

We thank the reviewer for highlighting this concern which we have fully considered. We refer the reviewer to answers made to reviewer 1 comment number 2 (*Phylogenetic reconstruction*) where we address these points.

3. While I appreciate the author's conclusion that the use of CCF is essential rather than just using VAF as a proxy for CCF, this conclusion ought to be clarified. Those methods that use VAF typically only use mutations in diploid regions (e.g., AncesTree, CITUP, PhyloSub, and many others) and many others have developed approaches for correcting VAF to account for CNAs (e.g., Canopy, PhyloWGS, SPRUCE, and many others). It seems that the authors are wanting to claim this observation as a novel contribution of this work, but the field is well aware of this limitation. So, additional information on what the authors are hoping to conclude with this point would be helpful.

We thank the reviewer for highlighting this important observation. We entirely agree that the field is now aware of the limitations of using VAF alone as a proxy for CCF, and we also agree

that the aforementioned software would be suitable for such analyses. Our discussions cautioning against the use of VAF alone are specifically directed towards previous melanoma studies that lacked these approaches. Such studies have coloured the accepted dogma around the lack of heterogeneity in melanoma metastases and we believe such conclusions should be cautioned against. To clarify this context, we have referenced these specific studies here and moved the penultimate sentence of the discussion (cautioning against the use of VAF as a surrogate for CCF) into its proper context specifically in relation to these studies (p.20):

'It is therefore our impression that previous studies suggesting that melanoma metastases lack heterogeneity may have been confounded either by the use of VAF as a surrogate for CCF²⁵, or by the lack of power to separate subclones through single sample analyses⁴⁰ (rather than by the limits of resolution of targeted sequencing approaches). Our analyses should therefore serve as a cautionary tale in future phylogenetic analyses that still define trunk and branch mutations by the presence or absence of shared variants and that do not consider CCF calculations (integrating information from somatic VAFs with tumour purity and ploidy considerations).'

We agree that (albeit important), this cautionary warning is not the novel contribution of this work. The key message of our study is that CCF clustering across multiple dimensions revealed that metastatic melanomas are characterised by previously unrecognised levels of intra-tumoural heterogeneity. At the start of the discussion we make specific reference to one of the largest pan-cancer evolutionary studies, which used a similar algorithm (also accounting for copy number aberrations) to infer evolutionary relationships across multiple cancer types (<https://doi.org/10.1101/312041>). The analyses in melanoma, focussed on single-sample metastases, concluded that melanomas uniquely lacked heterogeneity (<https://doi.org/10.1101/312041>). We therefore highlight that our two-stepped approach, first subtracting the dominant truncal variants, then applying CCF clustering across multiple dimensions, was able to uncover strong evidence of intra-tumoural heterogeneity. This was further replicated (using the same approach) in 7 independent cases. This is the first study to our knowledge to uncover conclusive evidence of ITH in melanoma metastases and we hypothesise that this approach could uncover similar insights in other malignancies. We have therefore amended the penultimate sentence of the discussion (previously relating to the cautionary warning described above) to highlight this contribution (p.23):

'Our ability to detect distinct clonal lineages was greatly enhanced by leveraging the power from multiple samples and, for the first time, uncovers conclusive evidence of ITH in melanoma metastases. Future large-scale studies incorporating clonal analyses across multiple metastases will be required to further delineate how these tumours evolve, and provide insights into whether interrupting this process could contribute to patient management.'

We hope that this amended concluding statement helps to clarify the novel contribution of this work, which we believe will be of interest to the community.

4. The author's conclusion of polyclonal seeding, Supplementary Figures 2 and 4, are difficult to follow. This needs to be made more clear how these conclusions are being drawn.

We thank the reviewer for highlighting this important issue relating to better clarity in the explanations of polyclonal seeding, which we have addressed in both in the main text and in the supplementary figures. We refer the reviewer to our answers to reviewer 1 comment 4 (*Biogeographic inference, see subsection entitled 'Polyclonal seeding and sample level phylogenetic trees'*).

Minor Comments: 1. Sometimes the terminology being used is unclear. While the authors do a good job of initially defining truncal, clonal and subclonal, later use of terms such as non-truncal vs. subclonal can be confusing.

We thank the reviewer for kindly pointing this out and for highlighting the need for consistency in usage of these key terms (clonal/subclonal/truncal). We have now addressed these in the text, and refer the reviewer to the responses made to reviewer one's 4th minor point entitled "*p. 3: the authors define truncal mutations as being in "all samples from the same patient" but hereafter, truncal mutations seem to be those exclusive of all metastatic samples.*"

2. For the example on page 5 about the order of mutations and copy number gains, it should be made explicit that the mutation is only contained on 2 out of 3 chromosomes in the instance that the copy containing the mutations was the one duplicated, in the other instance only 1 out of 3 will contain the mutation. 2

Thank you for highlighting this. We have amended this sentence (p.5) to read:

'For example, a mutation that has occurred on a chromosome that is subsequently duplicated is carried by two out of three chromosomal copies, whereas a mutation that occurred after the gain is carried by one out of three copies.'

3. Page 5 contains an incomplete sentence, "Indeed, whole-genome duplication and other copy number aberrations..."

Please accept our apologies for this, the second half of this sentence must have been inadvertently deleted during submission and did not occur in the preprint version [<https://doi.org/10.1101/848390>]. We have ensured that the full sentence is present and correct, which reads: *'Indeed, whole-genome duplication and other copy number aberrations vary across melanoma metastases from the same patient, evolutionary changes that may not be evident from the analysis of SNVs alone².'*

4. Figure 4 description - what statistical test was used to determine that Oxidative Phosphorylation was the most significant?

We thank the reviewer for highlighting this. The statistical calculations for the KEGG pathway enrichment analyses were undertaken within the *GSEAPreranked* (v6.0.10) software [PMID: 16199517]. In this software, the significance of an observed enrichment score (ES) is assessed by comparing the enrichment score with a set of scores computed with randomly assigned phenotypes, which generates a histogram of the corresponding null enrichment scores. The nominal P value is then calculated by using the positive (or negative) portion of the distribution corresponding to the sign of the observed enrichment score. To calculate the false discovery rate, the ES is normalized to account for the size of the gene set yielding a normalized enrichment score. The proportion of false positives is then calculated using the false discovery rate (FDR)⁴ corresponding to each normalized enrichment score (NES). The FDR is the estimated probability that a set with a given NES represents a false positive finding; it is computed by comparing the tails of the observed and null distributions for the NES [PMID: 16199517].

In the legend to Figure 4C (p.27), we have added a statement highlighting that oxidative phosphorylation is an MSigDB [PMID: 26771021] KEGG pathway gene-set, and have quoted the FDR-corrected p-value of for this gene set within this analysis (as well as beside the enrichment plot on Figure 4C). This sentence in the legend to Figure 4C (p.27) now reads:

*'Oxidative phosphorylation was the most statistically significant over expressed MSigDB KEGG pathway, enriched in both the 'brain versus normal tissue' and 'brain versus lung' comparisons (FDR corrected p-value<0.001, see **methods**), and has recently been linked to melanoma brain metastases in both human and murine analyses³⁴.*

In the gene expression analyses within the **methods** section (p.38), we have also explained the calculation of significance and false discovery rate as above, with this section now explaining:

'Hallmark gene sets were downloaded from the MSigDB database⁴⁸. Rank metric was calculated as the sign of log2-FCs calculated using the limma pipeline. The pipeline calculates an enrichment score (ES) that reflects the degree to which a gene set is overrepresented at the extremes (top or bottom) of the entire ranked list. The score is calculated by walking down the list, increasing a running-sum statistic when a gene in the set is encountered and decreasing it when a gene not in the set is encountered. The enrichment score is the maximum deviation from zero encountered in the walk and corresponds to a weighted Kolmogorov–Smirnov-like statistic⁵⁶. The significance of an observed enrichment score (ES) is assessed by comparing the enrichment score with a set of scores computed with randomly assigned phenotypes, which generates a histogram of the corresponding null enrichment scores. The nominal P value is then calculated by using the positive (or negative) portion of the distribution corresponding to the sign of the observed enrichment score. To calculate the false discovery rate, the ES is normalized to account for the size of the gene set yielding a normalized enrichment score. The proportion of false positives is then calculated using the false discovery rate (FDR)⁵⁷ corresponding to each normalized enrichment score (NES). The FDR is the estimated probability that a set with a given NES represents a false positive finding; it is computed by comparing the tails of the observed and null distributions for the NES⁵⁶.'

5. In the “whole-exome sequencing of multi-site metastases cases” section, the authors first say that Strelka was used to detect indels and later mention PINDEL, is this a typo or were both used?

Thank you for kindly spotting this typo, this is absolutely correct, only PINDEL was used to call somatic indels and we very much appreciate this. We have corrected this (p.35), as well as the accompanying reference.

6. Several methods have been developed recently to infer the evolutionary history of metastatic tumors. These should at least be cited in the Methods section. This includes, El-Kebir et al, Nature Genetics, 2018 and Reiter, J. G. et al., Nat. Commun., 2017.

We thank the reviewer for highlighting this which we agree would be useful to cite in the methods, (p.41):

‘The second step is to cluster SNVs based on their CCF by using the Bayesian Dirichlet process-based clustering in a multidimensional mode (ndDPClust (<https://github.com/Wedge-Oxford/dpclust>); as previously described⁴) implemented based on DPClust v2.2.8 (<https://github.com/Wedge-Oxford/dpclust>) to identify clonal and subclonal clusters across multiple samples of the same patient (other algorithms including that developed by El-Kebir et al⁶³ could also be used to infer the evolutionary history of multiple metastatic tumours).’

REVIEWER COMMENTS<

Reviewer #1 (Remarks to the Author):

The authors have done a thorough job addressing my comments. Indeed, the main point here is that there is quite a bit of heterogeneity among metastases in melanoma patients, and appreciate the authors focusing on this message in the revised version. I have only a few comments to further clarify some aspects. In particular, I still believe that the authors might consider improving the migration analyses a bit. All things considered, it is a nice paper, congratulations.

* Major specific comments

1) Evolutionary jargon

I see that the authors have removed the term "branching evolution" and use instead "divergent evolution". I appreciate it, but I have to say that, in my opinion, the latter term does not apply either, as "divergent evolution" is used in organismal biology primarily for phenotypes. The point is that every new mutation increases divergence at the molecular level, but not necessarily at the phenotypic level. So, at the molecular level, one could say that evolution is always divergent, although sometimes we say "recently diverged lineages" in contrast to (more) "divergent lineages". I understand that these concepts are subtle, but I believe that authors do not need to coin specific "modes" of evolution (I see too much of this in the cancer literature, often ignoring that most tumor lineages are never sampled and that selection acts on phenotypes, not on genotypes). Life is more complex than that. Do not take me wrong, but we do not always need to "squeeze" terms and concepts; we can often use plain English. In this particular case, I would just say "(high rates of) lineage diversification", or "divergent lineages" for example.

Please find below my suggestions for the statements you highlighted.

Section Page Replaced with

=> My suggestion

Abstract 2 Through whole-genome sequencing of 13 melanoma metastases sampled at autopsy from a treatment naïve patient and by leveraging the analytical power of multi-sample analyses, we reveal that metastatic cells may follow a divergent pattern of evolution.

=>..., we reveal high rates of diversification among metastatic lineages.

Abstract 2 Multi-sample analyses from a further 7 patients confirmed that divergent evolution was pervasive, representing ...

=> Multi-sample analyses from a further seven patients confirmed that lineage diversification was pervasive...

Results 11 Reconstructing the phylogenetic tree based on the metastatic non-truncal mutation clusters uncovers divergent evolution.

=>... uncovers high rates of diversification.

Results 14 Multi-site clonality analyses from a further 7 patients uncovers pervasive evidence of divergent evolution across melanoma metastases

=>...evidence of high diversification across melanoma metastases

Results 14 We identified 2-10 distinct clusters per patient with clear evidence of divergent evolution across ...

=>... with clear evidence of high rates of diversification across ...

Results 14 Given that divergent evolution was detected in 6/7 cases ...

=> Given that high rates of diversification were observed in 6/7 cases ...

Discussion 20 We found evidence of divergent evolution across metastatic melanoma exomes from ...
=> We found high rates of diversification across metastatic...

Figure 3 legend 26 Multi-dimensional Dirichlet processing across metastases from a further 7 patients uncovers evidence of divergent evolution...
=> ... uncovers evidence of a high diversification rate ..

Also, referring to ITH in metastases sounds weird, as you want to express heterogeneity among the secondary tumors of a given patient, and ITH means *intra* tumor heterogeneity. Again, I believe plain English will help to make the message more precise.

Finally, and I know it is going to be difficult to convince you of this ;-), but the term "trunk" seems to me not totally right (but I recognized I might have used it sometimes, in going with the flow; my fault). In nature, the trunk of most trees does not end after the first visible branching point, which is how it is used in the cancer literature. More importantly, in evolutionary biology we already have a name for the branch leading to the MRCA of the ingroup: it is the "root" branch.

2) Mutational clusters

I really appreciate the efforts of the authors to show that DPClust can be trusted in the proposed scenario. In my opinion, the validation and the simulation clearly strengthen the manuscript.

3) Phylogenetic reconstruction

Again, I appreciate the effort of the authors to clarify the phylogenetic reconstruction algorithm employed. I would add in the text that this approach assumes an infinite-site model. Still, note that this algorithm as such does not assess phylogenetic confidence. I encourage the authors to write at some point software for this (or why not just use say tools like PhyloWGS?), including some bootstrap procedure to evaluate how much the data supports the proposed phylogenetic hypothesis.

4) Time

The corrections in this regard are fine. It is important not to confound branch length with time, forgetting that the mutation rate per time unit does not have to be constant, particularly in cancer.

5) Migration

The authors have also adopted a more conservative approach regarding the interpretation of the different phylogenetic analyses in a biogeographical context. I believe it makes more sense now. However, please let me say that the migration inferences are still "eye-balling". Yes, I agree that polyclonal seeding can be inferred, however, in my mind your procedure is a bit manual and contains some "glitches". Being precise, not all these sample-level subtrees exist within each sample as such. If clone F migrated from c/d/e to f, then clone D was never present in f, the lineage D-F never existed there, and F is not related to A in this sample (so the A-D and D-F branches never existed in f). In my opinion, it would be much clearer just to label locations on the whole tree. Please note that inferring migration from trees is an old topic in evolutionary biology (e.g., PMID: 2599370), and some of the simpler methods have already been readily adapted to cancer (MACHINA, PMID: 29700472), while more sophisticated probabilistic approaches can also be used in this context (PMID: 31723138). Still, why do not you use a more reproducible methodology to infer migration, like MACHINA (standard parsimony ancestral character reconstruction), for example?

* Minor specific comments

p.3. "In the metastatic context, dissemination of cells from multiple lineages may lead to admixture of cell populations within multiple sites, likely with different CCFs at each site."
=> a reader could interpret that you are suggesting that heterogeneity within a metastatic site can only originate from multiple colonizations, but obviously it could just be diversification after a single colonization event.

p3. "For example, if one cell population is ancestral to another, its CCF must be greater in at least one sample and greater than or equal to the CCF of the descendant cell population in all other samples."
=> if you assume an infinite-site model.

p.3. "It should be noted that by constructing trees from clusters of mutations we avoid the previously-reported inaccurate inferences arising from ..."
=> I would say "we avoid potentially inaccurate inferences arising from ..."

p.4. I believe one of the most reliable examples of complex migration histories, and in my opinion the most sophisticated in terms of phylogenetic methodology, also using inferred clones, is <https://www.nature.com/articles/s41467-019-12926-8>. It is our paper but I think it makes all sense to cite it here, indeed at the author's discretion.

p.5. "polyclonal seeding (defined as a sample harbouring subclonal mutations from 2 or more diverged clonal lineages, thus representing multiple seeding events by two or more genotypically distinct cells)"
=> assuming these divergent lineages do not share a common ancestor in that sample. The concept you are looking for is polyphyletic lineages. I guess you could say: "polyclonal seeding (defined as a sample harbouring subclonal mutations from two or more distinct lineages that do not share an immediate common ancestor, thus representing multiple seeding events by two or more clones)"

p.5. "On the other hand, targeted sequencing approaches might not enable the detection of the whole catalogue of mutations, particularly heterogeneous mutations present in a small percentage of tumour cells, which could lead to an underestimation of ITH."
=> By definition, targeted sequencing approaches cannot enable the detection of the whole catalogue of mutations. And, on the contrary, for the genomic regions studied, targeted sequencing will detect in fact more often rare mutations than WES or WGS, as depth will be usually larger. Unless you are targeting known SNVs, in which case there will be an ascertainment bias toward higher allele frequencies. I suggest to revise the statement.

p.11. "Reconstructing the phylogenetic tree based on the metastatic non-truncal mutation clusters uncovers divergent evolution"
=> I would say "Phylogenetic analysis of the metastatic mutation clusters reveals distinct lineages" (it is obvious that what is shared (truncal) cannot differentiate)

p.11. "we reconstructed the phylogenetic tree of disease evolution"
=> this statement sounds weird to me. We reconstruct *a* tree, and not *the* tree (as we ignore truth), and I am not exactly sure of what is the "tree of disease evolution". I suggest rewording.

p.13. "This suggests that the long trunk of the phylogenetic tree could have originated from this subclonal cluster within the primary tumour."
=> I assume you mean the "long *metastatic* trunk", although you could just say that this subclonal cluster within the primary tumour might correspond to the lineage that originated the

metastases.

p.35. "Illumia" should be "Illumina"

Supplementary Fig. 3 does not seem to be cited in the text

Congratulations for a fine job. You are welcome to contact me for clarifications.

David Posada (dposada@uvigo.es)

Reviewer #2 (Remarks to the Author):

I found the revised manuscript by Rabbie/Ansari-Pour to be responsive to my critiques, and I hope the other reviewers feel similarly. It was enlightening to read the comments from the other reviewers – the cancer genomics community would certainly benefit from more cross-dialogue with the phylogenetics research community. My overall assessment remains positive because this manuscript is a substantial improvement in scope and quality over similar papers that study the evolution of metastatic melanoma, and the main conclusions, which I believe to be justified, will be of significant interest to the melanoma research community.

Minor point:

In my last round of review, I was confused whether C and F were mislabeled in figure 2D. After reading the authors' explanation, my source of confusion was rooted in the fact that they use letters to denote both mutation clusters in the top of figure 2C as well as letters to denote pieces of tissue in the right of figure 2C. The authors may wish to clarify this.

Hunter Shain
Alan.shain@ucsf.edu

Reviewer #3 (Remarks to the Author):

The authors have done a detailed and thorough job of addressing my concerns from the previous version of the manuscript. I also feel that a number of the changes to address the first reviewers' comments have substantially strengthened the manuscript.

REVIEWER COMMENTS

Reviewer #1 (Remarks to the Author):

The authors have done a thorough job addressing my comments. Indeed, the main point here is that there is quite a bit of heterogeneity among metastases in melanoma patients, and appreciate the authors focusing on this message in the revised version. I have only a few comments to further clarify some aspects. In particular, I still believe that the authors might consider improving the migration analyses a bit. All things considered, it is a nice paper, congratulations.

We thank the reviewer for their kind words and appreciate their positive view on the quality of this work. We would like to sincerely thank them for their thoughtful reviews and important suggestions. We agree that their comments have helped us focus the message of the manuscript, as well strengthen the mutational clustering and phylogenetic analyses. We have fully considered the reviewer's additional suggestions regarding the migration analyses as below.

* Major specific comments

1) Evolutionary jargon

I see that the authors have removed the term “branching evolution” and use instead “divergent evolution”. I appreciate it, but I have to say that, in my opinion, the latter term does not apply either, as “divergent evolution” is used in organismal biology primarily for phenotypes. The point is that every new mutation increases divergence at the molecular level, but not necessarily at the phenotypic level. So, at the molecular level, one could say that evolution is always divergent, although sometimes we say “recently diverged lineages” in contrast to (more) “divergent lineages”. I understand that these concepts are subtle, but I believe that authors do not need to coin specific “modes” of evolution (I see too much of this in the cancer literature, often ignoring that most tumor lineages are never sampled and that selection acts on phenotypes, not on genotypes). Life is more complex than that. Do not take me wrong, but we do not always need to “squeeze”

terms and concepts; we can often use plain English. In this particular case, I would just say “(high rates of) lineage diversification”, or “divergent lineages” for example.

We thank the reviewer for their important comment regarding this terminology. We entirely agree that clear and unambiguous terminology using plain English is critical to convey our core message. We would like to thank the reviewer for these specific textual suggestions, all of which we have incorporated into the revised manuscript. The reviewer will note that we have used the term ‘high rates’ sparingly, in order to avoid over-claiming our findings.

Please find below my suggestions for the statements you highlighted.

Section Page Replaced with

=> My suggestion

Abstract 2 Through whole-genome sequencing of 13 melanoma metastases sampled at autopsy from a treatment naïve patient and by leveraging the analytical power of multi-sample analyses, we reveal that metastatic cells may follow a divergent pattern of evolution.

=>..., we reveal high rates of diversification among metastatic lineages.

Thank you for this suggestion, which we have used:

=> ... *we reveal evidence of diversification among metastatic lineages.*

Abstract 2 Multi-sample analyses from a further 7 patients confirmed that divergent evolution was pervasive, representing ...

=> Multi-sample analyses from a further seven patients confirmed that lineage diversification was pervasive...

Thank you for this suggestion, we have amended as suggested.

Results 11 Reconstructing the phylogenetic tree based on the metastatic non-truncal mutation clusters uncovers divergent evolution.

=>... uncovers high rates of diversification.

Thank you for this suggestion which we agree is a better definition. However, we have proceeded with your suggestion in ‘minor specific comments’ p.14 below, which we think captures this finding even more sharply.

⇒ *Phylogenetic analyses of metastatic mutation clusters uncovers distinct clonal lineages*

Results 14 Multi-site clonality analyses from a further 7 patients uncovers pervasive evidence of divergent evolution across melanoma metastases

⇒...evidence of high diversification across melanoma metastases

⇒ *Multi-site clonality analyses from a further seven patients uncovers pervasive evidence of lineage diversification across melanoma metastases*

Results 14 We identified 2-10 distinct clusters per patient with clear evidence of divergent evolution across ...

⇒... with clear evidence of high rates of diversification across ...

⇒ *We identified 2-10 distinct clusters per patient with clear evidence of lineage diversification across 6 out of 7 patients*

Results 14 Given that divergent evolution was detected in 6/7 cases ...

⇒ Given that high rates of diversification were observed in 6/7 cases ...

⇒ (p.14-15) *Given that lineage diversification was detected in 6 out of 7 cases...*

Discussion 20 We found evidence of divergent evolution across metastatic melanoma exomes from ...

⇒ We found high rates of diversification across metastatic...

⇒ *We found evidence of lineage diversification across metastatic melanoma exomes from a further 6 out of 7 patients...*

Figure 3 legend 26 Multi-dimensional Dirichlet processing across metastases from a further 7 patients uncovers evidence of divergent evolution...

⇒ ... uncovers evidence of a high diversification rate ..

⇒ *Multi-dimensional Dirichlet processing across metastases from a further 7 patients uncovers evidence of divergent lineages.*

In light of these comments regarding this term, we carefully resolved the final few textual references to “divergent evolution” as follows:

Results p.14: In order to assess whether divergent evolution was detectable in further cases, we undertook whole-exome sequencing (WES) of 19 melanoma metastases matched with germline blood samples from an additional 7 patients with metastatic melanoma who had consented to take part in the MelResist study.

=> In order to assess whether lineage diversification was detectable in further cases, we undertook whole-exome sequencing (WES) of 19 melanoma metastases matched with germline blood samples from an additional 7 patients with metastatic melanoma who had consented to take part in the MelResist study.

Results p.20: By harnessing the power of CCF calculations across 6 metastases from our acral melanoma patient, we identified divergent evolution.

=> By harnessing the power of CCF calculations across 6 metastases from our acral melanoma patient, we identified divergent lineages.

Results p.21: Our analyses support these findings, and show that divergent evolution is associated with both locoregional as well as more distant metastatic spread.

=> Our analyses support these findings, and show that lineage diversification is associated with both locoregional as well as more distant metastatic spread.

Results p.22: Using the same approach, we found further pervasive evidence of divergent evolution in whole-exome sequenced metastases obtained from 6 out of 7 additional melanoma patients, one of which was an acral melanoma, suggesting that this is independent of sequencing breadth or depth.

=> Using the same approach, we found further evidence of divergent lineages in whole-exome sequenced metastases obtained from 6 out of 7 additional melanoma patients, one of which was an acral melanoma, suggesting that this is independent of sequencing breadth or depth.

Fig. 2D legend p.25: We observed evidence of divergent evolution emanating from a truncal clone.

=> We observed evidence of divergent lineages emanating from a truncal clone.

Methods p.44 To validate the divergent evolution pattern observed in the index autopsy case, six simulations were undertaken.

=> *In order to further validate the lineage divergence observed in the index autopsy case, six simulations were undertaken.*

Also, referring to ITH in metastases sounds weird, as you want to express heterogeneity among the secondary tumors of a given patient, and ITH means *intra* tumor heterogeneity. Again, I believe plain English will help to make the message more precise.

We thank the reviewer for this helpful comment. We have made this clarified this definition within the text as follows (p.4):

The term ‘intra-tumour heterogeneity’ (ITH) has been previously used to refer to heterogeneity identified from single or multi-sampling of tissue from a primary tumour. In this paper we extend the definition of ITH to ‘intra-patient tumour heterogeneity’, using it to refer to the observation of variants within a tumour that are non-truncal, including variants that may be clonal within some individual samples.

Finally, and I know it is going to be difficult to convince you of this ;-), but the term “trunk” seems to me not totally right (but I recognized I might have used it sometimes, in going with the flow; my fault). In nature, the trunk of most trees does not end after the first visible branching point, which is how it is used in the cancer literature. More importantly, in evolutionary biology we already have a name for the branch leading to the MRCA of the ingroup: it is the “root” branch.

We thank the reviewer for this important comment regarding usage of the term root branch in evolutionary biology. We entirely understand and appreciate this issue, but also need to be mindful that this may be less commonly used by the cancer community where, as the reviewer acknowledges (including in some of their own work), the term “trunk” is more widely used. Notwithstanding this, we agree with the importance of using established terms from the core evolutionary biology literature, and as part of the introduction on page 4 we provide a

definition of the key evolutionary terms used in this context. Herein we have further added reference to the term root branch as follows:

Throughout this study, we refer to mutations (and mutation clusters) observed in all tumour cells within a sample as ‘clonal’, those found in a subset of tumour cells as ‘subclonal’ and those found clonally in all samples from the same patient as ‘truncal’. We note that the term ‘trunk’ is used here in the same sense as the term “root branch” in the phylogenetic literature.

We hope that this will go some way in connecting the terms more commonly used in the cancer genomics community with core evolutionary biology nomenclature.

2) Mutational clusters

I really appreciate the efforts of the authors to show that DPClust can be trusted in the proposed scenario. In my opinion, the validation and the simulation clearly strengthen the manuscript.

We thank the reviewer for their kind compliment. We entirely agree that the orthogonal validation and phylogenetic tree simulations have strengthened the mutational clustering and sincerely thank the reviewer for their valuable suggestions here.

3) Phylogenetic reconstruction

Again, I appreciate the effort of the authors to clarify the phylogenetic reconstruction algorithm employed. I would add in the text that this approach assumes an infinite-site model. Still, note that this algorithm as such does not assess phylogenetic confidence. I encourage the authors to write at some point software for this (or why not just use say tools like PhyloWGS?), including some bootstrap procedure to evaluate how much the data supports the proposed phylogenetic hypothesis.

We thank the reviewer for this compliment and for the opportunity to clarify the description of our phylogenetic reconstructions. We thank the reviewer for this important comment regarding the infinite-site assumption which we entirely agree should be explicitly mentioned. We have

discussed this in previous publications using ndDPClust and thank the reviewer for prompting us to include it here, which we have done along with the primary reference:

Methods (Analysis of intra-patient tumour heterogeneity (ITH) and phylogenetic tree reconstruction) p.41:

Specifically, we applied the previously reported ‘sum’ and ‘crossing’ rules⁶⁶. Briefly, the sum rule operates upon the premise that if the CCFs of 2 mutation clusters in any sample add up to more than the CCF of their shared ancestral cluster, they must be collinear. The crossing rule states that if 2 mutation clusters B and C are descendants of mutation cluster A, and if cluster B has higher CCF than cluster C in one sample and cluster C has higher CCF than cluster B in another sample, clusters B and C must be branching. Any mutation cluster that violates these two principles is likely to be an artefact and thus removed from tree reconstruction. It should be noted that the sum rule and crossing rule only strictly apply when the infinite-sites model is assumed. This model states that each mutation only occurs once during the lifetime of a tumour and never reverts to normal⁶.

We thank the reviewer for their important comment regarding phylogenetic confidence. The first thing we would say is that although the sum and crossing rules do not generally restrict the possible trees to a single candidate, if we consider the mutational clustering presented in this study, then the hierarchical ordering of clusters only permits one phylogenetic tree solution. This is because all of our cases possessed one mutation cluster that was clonal in a subset of samples and another that was clonal in a complementing subset of samples. We have highlighted this in the methods (analysis of intra-patient tumour heterogeneity (ITH) and phylogenetic tree reconstruction) p.43.

The comment regarding building software to support the phylogenetic hypothesis is timely, as Dr David Wedge has recently taken on a PhD student whose project aims to extend ndDPClust to construct phylogenetic trees. We look forward to reporting on this in due course.

4) Time

The corrections in this regard are fine. It is important not to confound branch length with time,

forgetting that the mutation rate per time unit does not have to be constant, particularly in cancer.

We thank the reviewer for highlighting the isolated references to timing in the discussion. We agree that these corrections are more accurate and help focus on our key finding, that of previously unreported heterogeneity in melanoma metastases.

5) Migration

The authors have also adopted a more conservative approach regarding the interpretation of the different phylogenetic analyses in a biogeographical context. I believe it makes more sense now. However, please let me say that the migration inferences are still “eye-balling”. Yes, I agree that polyclonal seeding can be inferred, however, in my mind your procedure is a bit manual and contains some “glitches”. Being precise, not all these sample-level subtrees exist within each sample as such. If clone F migrated from c/d/e to f, then clone D was never present in f, the lineage D-F never existed there, and F is not related to A in this sample (so the A-D and D-F branches never existed in f). In my opinion, it would be much clearer just to label locations on the whole tree.

We thank the reviewer for highlighting this issue. We completely agree with the logic presented here and we apologise for inadvertently causing this confusion in Supplementary Figure 2. As they have non-identical CCFs in each sample, we know that D and F have separately migrated, providing evidence for polyclonal seeding. Similarly, A and B have separately migrated to or from metastatic sites PD38258f and PD38258o. However, as the direction of travel and the immediate source of these subclones is uncertain, we have removed the arrows from Supplementary Figure 2, instead drawing ovals to indicate those subclones that are evidence for polyclonal seeding. We have now updated the Figure considerably and believe it now provides a convenient and vivid visualisation for readers of the two main polyseeding events without the need to work out such events from the whole tree, which might not be straightforward for some readers. See also similar updates to Supplementary Figure 4 and associated legends.

Please note that inferring migration from trees is an old topic in evolutionary biology (e.g., PMID: 2599370), and some of the simpler methods have already been readily adapted to cancer (MACHINA, PMID:29700472), while more sophisticated probabilistic approaches can also be used in this context (PMID: 31723138). Still, why do not you use a more reproducible methodology to infer migration, like MACHINA (standard parsimony ancestral character reconstruction), for example?

We thank the reviewer for highlighting other approaches to inferring migration histories, specifically the MACHINA algorithm, which we agree provides a reproducible platform to infer migration histories in metastatic tumours. Through personal correspondence with the authors of MACHINA, we have learnt that this algorithm relies on having data from the matched primary tumour. Unfortunately given that whole-genome sequencing could not be undertaken in our primary tumour sample, we regret that this was not feasible.

Notwithstanding this, we do agree that this is a suitable algorithm to use in this context, and have highlighted its value within the methods:

(p.41-42) Analysis of Intra-patient tumour heterogeneity (ITH) and phylogenetic tree reconstruction:

The second step is to cluster SNVs based on their CCF by using the Bayesian Dirichlet process-based clustering in a multidimensional mode (ndDPClust (<https://github.com/Wedge-Oxford/dpclust>); as previously described⁴) implemented based on DPCLust v2.2.8 (<https://github.com/Wedge-Oxford/dpclust>) to identify clonal and subclonal clusters across multiple samples of the same patient. Other algorithms including that developed by El-Kebir et al⁶⁵ could also be used to infer the evolutionary history of multiple metastatic tumours (see Alves et al¹⁵). However, this requires equivalent data from the matched primary tumour, which was not feasible in this case.

* Minor specific comments

p.3. "In the metastatic context, dissemination of cells from multiple lineages may lead to

admixture of cell populations within multiple sites, likely with different CCFs at each site.”

=> a reader could interpret that you are suggesting that heterogeneity within a metastatic site can only originate from multiple colonizations, but obviously it could just be diversification after a single colonization event.

The intention of this sentence was to emphasise that subclones may be spread across multiple sites by polyclonal seeding, not that dissemination of tumour cells was the only source of heterogeneity. To clarify this, we have reworded this sentence as follows:

(p.3) In the metastatic context, dissemination of cells from multiple lineages may cause admixtures of cell populations to spread between different metastases, likely with different CCFs at each site.

p3. “For example, if one cell population is ancestral to another, its CCF must be greater in at least one sample and greater than or equal to the CCF of the descendant cell population in all other samples.”

=> if you assume an infinite-site model.

We thank the reviewer for this important comment which we agree should be mentioned (as above) and have added here again along with the primary reference as below:

Background p.3:

For example, if one cell population is ancestral to another, its CCF must be greater in at least one sample and greater than or equal to the CCF of the descendant cell population in all other samples, when assuming the infinite-sites model⁶. It should be noted that by constructing trees from clusters of mutations we avoid potentially inaccurate inferences arising from the construction of sample trees, when samples are an admixture of cells from multiple lineages⁷.

p.3. “It should be noted that by constructing trees from clusters of mutations we avoid the previously-reported inaccurate inferences arising from ...”

=> I would say “we avoid potentially inaccurate inferences arising from ...”

We thank the reviewer for this suggestion, which we agree is better suited to this context and have amended as suggested (p.3-4).

p.4. I believe one of the most reliable examples of complex migration histories, and in my opinion the most sophisticated in terms of phylogenetic methodology, also using inferred clones, is <https://www.nature.com/articles/s41467-019-12926-8>. [nature.com] It is our paper but I think it makes all sense to cite it here, indeed at the author's discretion.

We agree that this is an important and relevant paper to cite, particularly in the context of our discussions around inferring complex migration histories from multiple samples. We have cited this paper as follows.

Introduction (p.4):

Moreover, joint analysis of CCFs across multiple samples enables the identification of complex intermixtures of cell populations spread across multiple samples from a primary tumour as well as complex patterns of tumour cell metastasis⁸⁻¹⁴. Other approaches harnessing sophisticated biogeographic models to reconstruct clonal relationships across multiple samples have also provided detailed spatio-temporal insights into tumour evolution¹⁵.

p.5. "polyclonal seeding (defined as a sample harbouring subclonal mutations from 2 or more diverged clonal lineages, thus representing multiple seeding events by two or more genotypically distinct cells)"

=> assuming these divergent lineages do not share a common ancestor in that sample. The concept you are looking for is polyphyletic lineages. I guess you could say: "polyclonal seeding (defined as a sample harbouring subclonal mutations from two or more distinct lineages that do not share an immediate common ancestor, thus representing multiple seeding events by two or more clones)"

We thank the reviewer for pointing out this inaccurate wording. Of the samples that have evidence for polyclonal seeding, only one of them (sample PD38258f in Supplementary Figure 2) has evidence for polyphyletic seeding. Samples PD38258c, PD38258d, PD38258e and

PD38258o, however, have evidence of seeding from 2 subclones in an ancestral-descendant relationship. We would therefore prefer not to restrict our definition of polyclonal seeding to seeding of subclones from separate ‘phyla’. As our previous reference to ‘diverged clonal lineages’ was misleading, we have now reworded our definition of polyclonal seeding as follows (p.5):

‘defined as a sample harbouring subclonal mutations from 2 or more clonal lineages each of which is also found in another tumour site’

p.5. “On the other hand, targeted sequencing approaches might not enable the detection of the whole catalogue of mutations, particularly heterogeneous mutations present in a small percentage of tumour cells, which could lead to an underestimation of ITH.”

=> By definition, targeted sequencing approaches cannot enable the detection of the whole catalogue of mutations. And, on the contrary, for the genomic regions studied, targeted sequencing will detect in fact more often rare mutations than WES or WGS, as depth will be usually larger. Unless you are targeting known SNVs, in which case there will be an ascertainment bias toward higher allele frequencies. I suggest to revise the statement.

We thank the reviewer for spotting this statement which we agree needed revision. We have clarified this as below, which is more accurate and also fits much better with the preceding sentence in the paragraph (p.5-6).

Multi-site sequencing studies in melanoma have thus far been based on a small number of single nucleotide variants (SNVs) falling in coding exons, with gene panels focussed on SNVs in known cancer genes^{17,19,23-26}. While the high depth of sequencing used in these studies enables the detection of rare variants, the number of variants detected will be orders of magnitude lower than that from whole genome sequencing (WGS) and some clonal lineages may therefore go undetected.

The following sentence outlining the importance of accounting for copy number alterations in the estimation of clonal frequency now also links with this statement highlighting the

limitations of targeted sequencing approaches. We thank the reviewer for this helpful suggestion.

p.11. “Reconstructing the phylogenetic tree based on the metastatic non-truncal mutation clusters uncovers divergent evolution”

=> I would say “Phylogenetic analysis of the metastatic mutation clusters reveals distinct lineages” (it is obvious that what is shared (truncal) cannot differentiate)

We thank the reviewer for this suggestion, we have amended as suggested and agree this captures our key finding in a more poignant way.

p.11. “we reconstructed the phylogenetic tree of disease evolution”

=> this statement sounds weird to me. We reconstruct *a* tree, and not *the* tree (as we ignore truth), and I am not exactly sure of what is the “tree of disease evolution”. I suggest rewording.

We thank the reviewer for this suggestion, we have amended this and provided a clearer statement as follows:

*Assessing the distribution of these clusters as well as the CCF distribution within each cluster across the metastases, we were able to reconstruct a phylogenetic tree (see **Methods** for further details).*

p.13. “This suggests that the long trunk of the phylogenetic tree could have originated from this subclonal cluster within the primary tumour.”

=> I assume you mean the “long *metastatic* trunk”, although you could just say that this subclonal cluster within the primary tumour might correspond to the lineage that originated the metastases.

We thank the reviewer for this helpful suggestion which certainly provides a clearer description. We have amended this sentence as follows:

This subclonal cluster within the primary tumour might correspond to the lineage that originated the metastases.

p.35. "Illumia" should be "Illumina"

We thank the reviewer for kindly identifying this typo which we have corrected in the text.

Supplementary Fig. 3 does not seem to be cited in the text

We thank the reviewer for pointing this out, which also represents a typo. In the results section (p.14), the sentence:

We identified 2-10 distinct clusters per patient with clear evidence of lineage diversification across 6 out of 7 patients, evidenced by the presence of mutation clusters in mutually exclusive subsets of samples (Supplementary Fig. 2).

Should have rather referred to **Supplementary Fig. 3**, which shows clonal mutation clusters in mutually exclusive samples from MultiSite_WES_Patient1, a finding which was replicated across all the MultiSite_WES_Patients. We thank the reviewer for kindly identifying this typo which we have corrected.

The comment prompted us to re-review all the textual figure references which were otherwise all correct.

Congratulations for a fine job. You are welcome to contact me for clarifications.

We thank the reviewer again for their outstanding and thoughtful review which has greatly benefited this manuscript, as summarised by reviewer two below.

David Posada (dposada@uvigo.es)

Reviewer #2 (Remarks to the Author):

I found the revised manuscript by Rabbie/Ansari-Pour to be responsive to my critiques, and I hope the other reviewers feel similarly. It was enlightening to read the comments from the other reviewers – the cancer genomics community would certainly benefit from more cross-dialogue with the phylogenetics research community. My overall assessment remains positive because this manuscript is a substantial improvement in scope and quality over similar papers that study the evolution of metastatic melanoma, and the main conclusions, which I believe to be justified, will be of significant interest to the melanoma research community.

We thank the reviewer for their helpful summary and kind compliments. We entirely agree with the benefit we have accrued from cross-dialogue with the phylogenetics research community, and agree this could be incredibly beneficial to the wider cancer community. We thank the reviewer for their important compliment regarding the scope and quality of this work. We consider this reviewer to be one of the foremost experts in the field and enormously appreciate these comments regarding the potential interest to the melanoma research community.

Minor point:

In my last round of review, I was confused whether C and F were mislabeled in figure 2D. After reading the authors' explanation, my source of confusion was rooted in the fact that they use letters to denote both mutation clusters in the top of figure 2C as well as letters to denote pieces of tissue in the right of figure 2C. The authors may wish to clarify this.

We thank the reviewer for this excellent suggestion and apologise we hadn't spotted this in the original comment. We have amended the sample labels for figure 2C, replacing the lower-case lettering with the full sample ID from the WGS analysis (which includes the patient prefix 'PD38258' followed by the sample prefix, the latter represented by lower-case lettering). These full sample IDs now more clearly connect with the axes labels shown on the representative density plots in Figures 2 A & B. In addition, the reader will now also be able to track these sample IDs to Supplementary Table 1 (including all the corresponding clinical details) as well as the relevant sequencing data repository (see Data and software availability p.47, <https://ega-archive.org/studies/EGAS00001001348>). Following on from this helpful suggestion, we have

replaced the sample lettering with full sample IDs in all the remaining figures, including Supplementary Figures 2, 3 and 5. This again will allow for easier tracking of sample IDs to the corresponding clinical and sequencing data (in Supplementary tables and the EGA archive respectively). We thank the reviewer for this very helpful suggestion.

Hunter Shain

Alan.shain@ucsf.edu

Reviewer #3 (Remarks to the Author):

The authors have done a detailed and thorough job of addressing my concerns from the previous version of the manuscript. I also feel that a number of the changes to address the first reviewers' comments have substantially strengthened the manuscript.

We thank the reviewer for their kind compliment. We are extremely grateful to this reviewer and to reviewer 1 for raising very similar comments and suggestions, which we entirely agree have greatly strengthened the quality and clarity of this manuscript.